# The effect of obliquity on temperature in subduction zones: insights from 3D numerical modeling.

Alexis Plunder[1,a], Cédric Thieulot[1], and Douwe J. J. van Hinsbergen[1]

[1]Department of Earth Sciences, Utrecht University, The Netherlands
[a]Now at Sorbonne Université, CNRS-INSU, Institut des Sciences de la Terre Paris, ISTeP UMR 7193, F-75005 Paris, France

*Correspondence to:* A. Plunder, (alex.plunder@gmail.com)

**Abstract.** The geotherm in subduction zones is thought to vary as a function of the subduction rate and the age of the subducting lithosphere. Along a single subduction zone the rate of subduction may strongly vary due to changes in the angle between the trench and the plate convergence vector, i.e. the subduction obliquity, due to trench curvature. We currently observe such curvature in e.g. the Marianas, Chile, and the Aleutian trenches. Recently, strong along-strike variations in subduction obliquity were proposed to have cause a major temperature contrast between Cretaceous geological records of Western and Central Turkey. We here test whether first-order temperature variation in subduction zone may be caused by variation of the trench geometry using simple thermo-kinematic finite element 3D numerical models. We prescribe the trench geometry by means of a simple mathematical function and compute the mantle flow in the mantle wedge by solving the equation of mass and momentum conservation. We then solve the energy conservation equation until steady-state is reached. We analyze the results (i) in terms of mantle wedge flow with emphasis on the trench-parallel component, (ii) in terms of temperature along the plate interface by means of maps and depths-temperature path at the interface. In our experiments, the effect of the trench curvature on the geotherm is substantial. A small obliquity yields a small but not negligible trench parallel mantle flow leading to differences of 30°C along strike of the model. Advected heat causes such temperature variations (linked to the magnitude of the trench parallel component of velocity). With increasing obliquity, the trench parallel component of the velocity consequently increases and the temperature variation reaches 200°C along strike. Finally, we discuss the implication of our simulations for the ubiquitous oblique systems that are observed on Earth and the limitation of our modeling approach. Lateral variations in plate sinking rate associated with curvature will further enhance this temperature contrast. We conclude that the synchronous metamorphic temperature contrast between Central and Western Turkey may well have resulted from reconstructed major variations in subduction obliquity.

## 1  Introduction

Oceanic subduction and continental collision zones represent approximately 55 000 kilometers of converging plate boundaries on Earth today. They are primarily associated with arc magmatism and seismicity, which in turn are mainly a response to the thermal structure and geotherm of a subduction zone. Numerous studies using 2D high resolution numerical models have addressed the effect of temperature in subduction zones and its link to the coupling of the subduction interface (Wada and

Wang, 2009) and related seismicity (Kirby et al., 1996; Peacock and Wang, 1999; Hacker et al., 2003b), as well as the release of fluids (van Keken et al., 2011; Wada et al., 2012), and the associated generation of melt (Gorczyk et al., 2007; Bouilhol et al., 2015). Temperature distributions in subduction zones are thought to vary primarily as a function of the subduction rate and the age of the subducting lithosphere, with lower subduction rates and younger lithosphere tend to increase temperatures

at the subduction interface (Kirby et al., 1991; Peacock and Wang, 1999; van Keken et al., 2011). The geotherm is then mainly controlled by trench perpendicular flow (poloidal) with presumably little variation along-strike. This poloidal flow allows the transport of heat by means of advection (see below).

     Real subduction zones, however, tend be curved, i.e. trench strike varies laterally and the angle between the absolute plate motion at the trench and trench strike – the subduction obliquity – thus change along-strike. In fact, some degree of oblique

subduction is the rule rather than the exception, both in todays snapshot of plate tectonics (e.g. Fig. 1 and Bird (2003)) as well as in the geological past (Stampfli and Borel, 2002), e.g. in the Mediterranean region (e.g. van Hinsbergen et al., 2016; Menant et al., 2016), the South-American system (Vérard et al., 2012; Schepers et al., 2017) or the western North American margin (Johnston, 2001; Liu et al., 2008).

Lateral variations in subduction obliquity may conceptually influence the temperature at the subduction interface in two ways. First, oblique subduction adds a component of horizontal relative motion between slab and mantle wedge - toroidal flow - to the poloidal flow in the mantle wedge, which may influence heat advection. Second, higher subduction obliquity leads to a lower net subduction rate. Intuitively, this may suggest that increasing subduction obliquity may be associated with higher temperatures at the subduction interface. Investigating the effect of trench curvature on along-strike variations of

temperature at the plate interface may thus help to explain along-strike variations in e.g. generation of magma, or seismicity along subduction zones, or to help the reconciliation of contrasting metamorphic records with kinematic reconstructions. Few studies were conducted on the effect of obliquity/geometry on the geotherm of subduction zone (e.g. Ji and Yoshioka, 2015) and most studies mainly focused on mantle flow patterns (Honda and Yoshida, 2005; Kneller and van Keken, 2008; Jadamec and Billen, 2010, 2012; Bengtson and van Keken, 2012; Morishige and van Keken, 2014; Wada et al., 2015)

In this paper, we aim to study the effect of the trench curvature on along-strike temperature distribution changes in subduction zones. In particular, we aimed to test a recent hypothesis that a major, more than 300°C along-strike contrast in subduction zone temperature concluded from the geology of Turkey resulted from a corresponding major change in trench strike (see next section; van Hinsbergen et al., 2016). To this end, a simple 3D thermo-kinematic numerical setup was designed and computed using the finite element code ELEFANT (Thieulot, 2014; Lavecchia et al., 2017). Below, we review selected present-

30    day subduction zones and their geometric characteristics as basis for our numerical model setting. Then, we summarize the rationale behind the hypothesis of van Hinsbergen et al. (2016) relating lateral variations in metamorphic grade recognized in the geology of western and central Turkey to oblique subduction. After that, we provide the results from a series of 3D numerical experiments and discuss the limitations of our simple experiments. We evaluate the implications of slab shape or trench geometry on the thermal regime of subduction zone and finally, compare the numerical results with the geological

examples of Turkey and the Franciscan complex.

## 2 Oblique subduction: present and past examples

Many present-day subduction zones show an along-strike variability in the angle between the absolute motion direction of the downgoing plate and the trench. Fig. 1 shows that a majority of 100s to >1000 km long subduction zones have concave (e.g. Marianas; Sunda-Burma), or convex (central South America, Northeast Japan) shapes. Some trenches contains as much as 90°curvature such that along the same trench, subduction may gradually (Aleutians, Sunda-Burma) or abruptly (southern Marianas, northern Lesser Antilles) change from near-orthogonal subduction to near-transform motion. The subduction rate along such curved subduction zones must change as a function of trench strike. This is best illustrated by the Aleutian trench (Fig. 1). In the eastern, NE-SW striking part of the trench, subduction is almost orthogonal, i.e. the plate motion of the downgoing Pacific plate is almost perpendicular to trench strike. In the western, NW-SE striking part of the Aleutian trench, there is almost no subduction and Pacific plate motion is almost parallel to the trench (e.g. Mccaffrey, 1992). This is also reflected in the westward decrease of the length of the Aleutian subducted slab (van der Meer et al., 2017). Consequently, the subduction rate along the Aleutian trench must gradually decrease from east to west with increasing subduction obliquity.

If subduction rate is a primary control on the geotherm (e.g. van Keken et al., 2011), then along-strike variation in obliquity, should logically lead to along-strike changes in temperature at subduction interfaces. However, determining how strong these lateral variations may be is difficult to estimate from present-day subduction zones due to the lack of proxy to record them. Plank et al. (2009) provided a method to estimate the temperature at the plate interface using melt-inclusions in arc volcanic rocks. Such data suggested along strike variations of temperature exist and can vary through time for example below Central America (Cooper et al., 2012). Better-constrained estimates for the temperature are available for paleo-subduction interfaces through studies of exhumed metamorphosed rocks in subduction-related orogens. These studies demonstrated that the thermal conditions in subduction zones varied through time (e.g. Agard et al., 2009; Plunder et al., 2015; Angiboust et al., 2016), but also along-strike. For instance, in the Franciscan complex of California (Wakabayashi and Dumitru, 2007), in the Sulawesi mélange in SE Asia (Parkinson, 1996), in the sub-ophiolitic mélanges of Guatemala versus Cuba (Garcia-Casco et al., 2007), (garnet)-amphibolites (high-temperature and mid-pressure condition) are coeval with eclogite or blueschist (low temperature and high-pressure condition) along-strike in the same subduction complex. Taking the pressure (simply assumed to represent depth) of metamorphism into account, these may suggest along-strike temperature differences of ca. 300°C. Less dramatic along strike temperature differences at similar depth of (ca. 100°C) have also been recorded in Miocene subduction-related metamorphic rocks of Crete, Greece (Jolivet et al., 2010).

An extreme case of along-strike coeval metamorphic temperature variation was reconstructed from the geological record of Turkey. There, two belts of metamorphosed continental rock known as the Tavşanlı zone and the Kırşehir block experienced coeval metamorphism at strongly contrasting grades during their underthrusting/subduction below oceanic lithosphere that is preserved as ophiolites (e.g. Boztuğ et al., 2009; Plunder et al., 2013; van Hinsbergen et al., 2016). Some of these ophiolites formed above the nascent subduction zone and are referred as of supra-subduction zone type (Pearce et al., 1984; Dilek et al.,

1999). They formed ∼5-10 Myr before climax metamorphism of the Kırşehir Block and Tavşanlı zone (van Hinsbergen et al., 2016, and references therein).

Both units were metamorphosed around 80-90 Ma (e.g. Whitney and Hamilton, 2004; Fornash et al., 2016; van Hinsbergen et al., 2016; Pourteau et al., 2018) within the same subduction system but under dramatically different metamorphic conditions. In the Tavşanlı zone peak metamorphic conditions were estimated to be around $24 \pm 2$ kbar and $500 \pm 50°C$ (Okay et al., 2002; Plunder et al., 2015), whereas peak metamorphism was estimated around $800 \pm 100°C$ and $8 \pm 1$ kbar in the Kırşehir block (Whitney and Hamilton, 2004; Lefebvre et al., 2015). This would suggest that at similar depths, an along-strike temperature variation of more than $500°C$ existed (i.e. ∼$200°C$ for the Tavşanlı zone at ∼25 km depths compared to $800°C$ for Kırşehir at the same depth). The paleogeographic transition between the Tavşanlı and Kırşehir blocks has been deformed during later continent-continent collision processes, but appears to be abrupt, presently within tens of kilometers (Fig. 1b).

Paleogeographic and kinematic reconstructions of Central and Western Anatolia (Lefebvre et al., 2013; van Hinsbergen et al., 2016; Gürer et al., 2016), suggest that the only major difference between the Tavşanlı and Kırşehir parts of the belt was the angle at which they were buried along the intra-oceanic trench below the oceanic lithosphere now found as ophiolites (Fig. 1b). Such reconstruction are constrained based on structural geology and paleomagnetism, and more importantly are independent from interpretations of the causes of the contrast in metamorphism. Subduction of the belt was driven by ∼NNE-SSW convergence between Africa and Eurasia. The Tavşanlı zone was proposed to have been buried by near-orthogonal subduction along an ∼E-W trending trench segment, whereas the Kırşehir block would have been subducted highly obliquely (Fig. 1b) along a N-S striking trench segment, which was tentatively proposed to explain the stark metamorphic contrast (van Hinsbergen et al., 2016). Inspired by these geological examples and the hypothesis derived from those, we here aim to perform numerical experiments to test whether, and to what extent, the reconstructed thermal variations may be explained by along-strike variation in subduction geometry.

## 3 Model setting

### 3.1 Background

With increasing quality of geophysical measurements and network density, today's tomographic images allow to observe the geometry variation of slab geometry with depths (van der Meer et al., 2017). Such variations of slab shape are observed below the strait of Gibraltar (Bezada et al., 2013), below Turkey (Biryol et al., 2011), below Japan (Zhao et al., 2012; Liu and Zhao, 2016), below the eastern Caribbean plate (Van Benthem et al., 2013) and in many other subduction zones, and are summarized in the SLAB1.0 model (Hayes et al., 2012). These complicated pictures of slab geometry allow us to make simple tests to study the possible effects of geometry on the mantle flow and on temperature in subduction zones, and especially at the subduction interface.

Previous 3D thermo-kinematic numerical modeling studies have shown that variation of the geometry of the subduction zone may affect mantle flow patterns and may help to explain seismic anisotropy observed in subduction systems (e.g. Kneller

and van Keken, 2007). Numerical models also suggested that the obliquity of subduction zones may have an effect on the temperature at the subduction interface (Bengtson and van Keken, 2012; Morishige and van Keken, 2014; Ji and Yoshioka, 2015) but did not explore the relationship of such effects with the geological record. These studies have primarily shown that mantle flow may be related to the geometry of the slab edges that lead to the development of toroidal cells (i.e. with trench

parallel material transport; Király et al., 2017; Schellart, 2017). Such trench-parallel, toroidal mantle flow has been proposed as a possible mechanism for differences in volcanic activity along subduction strike (Faccenna et al., 2010). Some mechanical studies have investigated the effect of trench geometry on the development of topography in the upper plate (e.g. Bonnardot et al., 2008). They also showed that plates bend in relation to the trench shape. Schellart et al. (2007) in their study showed that the shape of a slab is controlled by its original width and evolve in time. Similar studies show that dynamic subduction

systems develop 3D geometry with curvature as observed in nature, but in general such models are only mechanical and do not consider temperature (Pusok and Kaus, 2015; Király et al., 2017; Schellart, 2017), or the temperature pattern was not discussed in detail (Jadamec and Billen, 2010, 2012; Chertova et al., 2014; Haynie and Jadamec, 2017). Hence, in our study, we aim to test to what extent trench geometry influences the geotherm of a subduction zone.

## 3.2   Numerical rationale and methods

The pioneering works of Batchelor (1967) and McKenzie (1969) allowed to investigate the thermal state of subduction zones by providing an analytical solution in 2D, whereby corner flow (i.e. poloidal flow) is dominant. Following these works, many studies were conducted on the behavior of subduction zones using analytical solution (Tovish et al., 1978; England and Wilkins, 2004) or numerical approximations of corner flow, taking into account stress and temperature dependence of the material in the mantle wedge (e.g. van Keken et al., 2002, and references therein). However subduction and particularly the shape of slabs

is a 3D problem for which no simple analytical solution exists. To investigate the effect of obliquity on mantle flow and on the temperature at the plate interface, we designed a simple numerical setup using a reference model, and compute deviations from that reference for a set of models in which we vary trench shape. In addition, we briefly test the effect of subduction rate, subduction angle angle an the downgoing plate age on the thermal state of the plate contact for the reference model. For geological cases where a plate subducts with an along strike varying obliquity it is then possible to add up the effects of trench

geometry and subduction rate on mantle flow and therefore on the temperatures.

We used the finite element code ELEFANT (Thieulot, 2014; Lavecchia et al., 2017) to solve the mass, momentum and energy conservation equations in three-dimensions:

$$\boldsymbol{\nabla} \cdot \boldsymbol{v} = 0 \tag{1}$$

$$-\boldsymbol{\nabla} P + \boldsymbol{\nabla} \cdot (2\mu\dot{\varepsilon}) = 0 \tag{2}$$

$$\rho_0 C_p \left( \frac{\partial T}{\partial t} + \boldsymbol{v} \cdot \boldsymbol{\nabla} T \right) = \boldsymbol{\nabla} \cdot (k\boldsymbol{\nabla} T) \tag{3}$$

under the Boussinesq approximation with $\boldsymbol{v}$ the velocity, $P$ the dynamic pressure, $\mu$ the effective viscosity, $\dot{\varepsilon}$ the strain-rate tensor, $\rho$ the volumetric mass density, $C_p$ the specific heat, $T$ the absolute temperature, $t$ the time and $k$ the thermal conductivity. All parameter values are given in table 1. The domain consists of a non-deforming upper plate, a slab with a kinematically prescribed velocity and an isoviscous dynamic mantle wedge. All coefficients were assumed to be constant both in time and space so that the temperature has no effect on the solution of Eqs. (1,2). As a consequence, once Eqs. (1,2) have been solved for a given set of boundary conditions and geometry, the same velocity field $\boldsymbol{v}$ is used to solve Eq. (3). This allows for a substantial reduction of the computational time since the discretisation of the Stokes equations yields a saddle point problem that is known to be much more computationally demanding than the energy system (Donea and Huerta, 2003).

We designed our model to be at first order similar to our Anatolian case study (Fig. 1b) with several simplifications. Therefore we prescribed the velocity boundary conditions and geometry as shown in Fig. 2. They are summarized as follows: (i) a subduction rate of 40 mm.yr$^{-1}$ was imposed with a dip angle of $45°$ for the slab (Fig. 2), a combination that is reasonable considering present-day subduction zones (Syracuse et al., 2010) and that is similar to reconstructed Africa-Europe convergence rates around 90-80 Ma (Seton et al., 2012), and thus comparable to the Anatolian case study; (ii) the top 32 km of the mantle wedge was assumed to be rigid to mimic the mechanical behavior and the thickness of a 5 million year old crust and shallow lithosphere (i.e. the age of most ophiolite in Turkey at the time of the metamorphic contrasts), similar to the Anatolian case study; (iii) no in- or out-flow was allowed in the direction parallel to the trench ($v_y = 0$); (iv) no vertical movement was allowed in the rear of the modeling space.

The temperature at the surface was set to $0°$C. At the front and the rear of the domain, the temperature was computed using a half space cooling model that is in good agreement with various geophysical observations for oceanic lithosphere younger than 60 My (Turcotte, 1987). The age was set to 25 My for the subducting plate and as a 5 My old lithosphere for the upper plate. In the rear of the modeling space, the thermal state is prescribed until reaching the in/out flow transition at 100 km depth (Fig. 2). This in/out flow transition was set in order to allow the corner flow thermal structure to develop (van Keken et al., 2002; Currie et al., 2004; Wada et al., 2015).

The computational domain is discretized on a grid counting $65 \times 85 \times 65 = 359,125$ elements allowing a physical resolution of $2.3 \times 3 \times 2.3$ km in the $x$, $y$ and $z$ directions. In all calculations we used linear $Q_1 Q_1$ elements for velocity, pressure and temperature. Since equal-order-interpolation for the velocity-pressure pair is known to yield an unstable mixed finite element formulation we used the stabilization method of Dohrmann and Bochev (2004) that was previously successfully implemented in other geodynamic models (Stadler et al., 2010; Burstedde et al., 2013). We used a preconditioned Conjugate Gradient method to solve the Schur complement of the Stokes system. Inner solves are carried out with the direct solver MUMPS (Amestoy et al., 2001, 2006) while a GMRES solver was used for the energy equation finite element matrix. All the calculation procedure are explained in Thieulot (2011, 2014). The simulations are run until the temperature pattern in the slab is not mainly driven by the advection term of the energy equation (Eq. 3). We run the calculations until steady state is reached (ca. 15-20 Ma depending

on the simulation with a time step of about 5000 years) on a Desktop machine using a single processor. Each model took about one to two hours to compute.

## 3.3 Geometry of the models

At the beginning of each simulation the grid was built as a Cartesian box and then deformed to conform to the required curved
geometry and boundary conditions imposed. The position $x_t$ of the trench as a function of the $y$ coordinate is prescribed by means of a sine or an arctangent function:

$$x_t(y) = x + A \left[ 1 - \left( \sin \frac{y - \frac{Ly}{2}}{L_y} \pi \right)^{2\beta} \right] \tag{4}$$

$$x_t(y) = x + A \left[ \arctan \left( \gamma \left( \frac{y}{L_y} - \frac{1}{2} \right) \right) \right] \tag{5}$$

where A is the amplitude of the curvature and $\beta$ and $\gamma$ are parameters controlling its shape. The angle between the trench and
the direction of convergence, parallel to $x$, is called $\theta$ and varies with $y$ (Fig. 2).

## 4   Results

The models are named after the parameters controlling the shape of their trench (A, $\beta$ or $\gamma$; see Eqs. 4 or 5; e.g. SINA_$\beta$). Model SIN20_1 will be described in detail and serves as a reference against which other runs are compared. This model has a sinusoidal shape with an amplitude of 20 km and a $\beta$ value of 1. For all the convex models, the along strike temperature
variation at 75 km depths ($\Delta T_{75km}$) and maximum obliquity ($\theta_{max}$) is reported in Tables 2 and 3. For all the experiments the value of $\theta_{max}$ is indicated on the figure describing the experiment. We provide movies of all models in supplementary materials.

## 4.1   Boundary conditions in the rear of the model

Two different types of boundary conditions were investigated for the rear of the box: (i) the in/out flow is allowed only in
the $x$ direction (with $v_y = 0, v_z = 0$) (ii) the in/out flow allowed in the $x$ and $y$ direction (with only $v_z = 0$). At a depth of 60 kilometers the difference in the mantle wedge flow is minimal when the subducting slab is far from the rear of the box. There, the maximum $v_y$ value is 1.41 mm/yr and 1.54 mm/yr with the (i) and (ii) boundary condition respectively, i.e. a $\sim 8\%$ difference. At depths of 75 and 90 kilometers and closer to the rear of the modeling space this difference increases to 13% and 26%, respectively. In what follows, we chose the second approach were flow comes in horizontally from the top right boundary
and flows out at the bottom boundary, conserving the mass in the system to be more realistic.

## 4.2 Description of the reference experiment: SIN20_1

### 4.2.1 Mantle wedge flow

The 3D velocity pattern of the mantle wedge is shown using streamlines colored with trench-parallel velocity $v_y$ (Fig. 3) and cross sections at different depth intervals (60, 75 and 90 km). In our computation we observe that the shape of the box influences the mantle wedge flow (i.e. the geometry of the trench and slab). Some inflow of mantle comes from the backarc region and is dragged at depth due to the viscous coupling with the subducting slab. This is especially true in the middle and edges of the box where the obliquity angle is null. This is shown on the top and rear view of the model (Fig. 3a,b) where the mantle flow on both sides and in the middle does not depict a trench parallel component. In the inflow area the trench-parallel component of the velocity is close to zero (0.05 mm/yr). The material is drawn in in the arc region and transported linearly towards the subduction area. In the narrower part of the wedge, the trench-parallel component of the velocity increases up to 2.9 mm/yr reaching there a maximum when the obliquity angle is maximum (at 64 km from the edge on the reference model SIN20_1; Fig. 3a, cross section). It almost corresponds to the location where the flow reverses and where the velocity is of ∼5 mm/yr. Along the subducting plate, the mantle flow follows the interface. The streamlines are affected by some trench-parallel flow around the maximum curvature region (y = 64 km and y = 128 km; Fig. 3a,b). There, the $y$ component of the velocity reach at maximum 2.9 mm/yr and is considered as almost negligible compared to a magnitude velocity up to 30.0 mm/yr ($v_y$ = ∼9.6% of $v$). In the outflow area the average magnitude velocity is 12.6 mm/yr. It is mostly composed of the $x$ component ($\overline{v}_x$ = 12.5 mm/yr) and the $y$ component corresponds to ∼12% on average the total velocity ($\overline{v}_y$ = 1.20 mm/yr). In the wedge, the $y$ component of the velocity is maximum at a position equivalent to 1/4 of the box size (i.e. where $\theta \to \theta_{max}$; Fig. 3a,b). Its value is 1.54 mm/yr at 60 km depth, 1.42 mm/yr at 75 km and 0.94 mm/yr at 90 km depth. This corresponds to ∼12%, ∼16% and ∼17% of the magnitude velocity $v_{mag} = |v|$ at the same location respectively (Fig. 3d,e,f and Table 2).

In summary, a trench parallel flow develops in the mantle wedge in relation with the shape of the trench. The trench parallel component of the flow cancels at the center of the model and adds up to the along plate interface flow to transport more material at depth.

### 4.2.2 The thermal structure

The thermal structure is presented in a 3D view and as map views across the mantle wedge at different depths (Fig. 3). In addition, depth-temperature paths measured along the plate interface are provided but only shown for one half of the model because they are symmetric. As discussed before, the velocity field converges towards the center of the modeling space (Fig. 3). Due to advection it appears logical that the 450°C isotherm is deflected downward in the center of the modeling space (i.e. where the trench parallel velocity becomes zero; Fig. 3c). As a consequence the thermal regime of the subduction zone is different along strike and is cooler in the middle of the domain.

This is also well illustrated with the depth-temperature path along the interface (Fig. 4a). Fig. 4a shows that the path in the center of the model is the coldest and that paths where theta tends to a maximum are the warmest (paths 1 to 4 being within 1 degree at a similar depth; see zoom on Fig. 4). For this reference model the along-strike variation of temperature at the

subduction interface is 33°C at a depth of 75 km (See inset on Fig. 4a). This variation becomes smaller at shallower depth when the subduction interface gets closer to the fixed overriding plate. This region namely the cold nose is even sometimes considered as fixed (e.g. van Keken et al., 2002).

### 4.2.3  The effect of slab age, slab velocity and the subduction angle on the thermal structure

A set of additional experiments with varying slab age, subduction rates and subduction angle was also calculated. We tested the thermal structure at steady state for subducting slab with thermal ages of 50, 75 and 100 My in addition to the reference case where the thermal age is 25 My. The results are shown on Fig. 4b. As expected, the thermal regime decreases with the increasing age of the downgoing plate. The temperature spread along strike is not influenced much by the age of the slab and gives temperature difference of ∼33°C along strike (Table 2).

With velocities of 20 and 70 mm/yr for the reference concave model (and compared to the 40 mm/yr velocity in the reference experiment) the temperature spread along strike show slight variations (Table 2). With 20 mm/yr, the geotherm of the subduction zone gets about 50°C warmer (as represented by the red depth-temperature path on Fig. 4c with a difference along strike of 34°C (Table 2). With 70 mm/yr the global geotherm gets 50°C colder (blue depth-temperature path on Fig. 4c) with a difference along strike of 31°C (Table 2). The temperature difference along strike is for the three cases largely within the uncertainties of our calculations and we considered that the difference has no proper significance and we consider a difference of ca. 30°in each models.

With a subduction angle of 30°($\alpha$ on Fig. 2), compared to the 45°angle in our reference experience, the temperature spread slightly increase along strike and reaches ca. ∼50°C as shown by the depth-temperature curves on fig 4c. Again with varying velocities the general thermal regime in decreasing with increasing subduction rate (Fig. 4d). All results are summarized in table 2.

## 4.3  Summary of the other experiments

### 4.3.1  Convex and concave models

**Convex models:** The maximum value of $\theta$ is measured either at ∼ 42 or 64 km corresponding to the so-called inflection point (i.e. where the second derivative of eq. 4 equals zero). As in the reference experiment, the mantle wedge flow shows a trench parallel velocity with a increasing value in the region where the obliquity is the highest. The maximum trench parallel velocity is reported in table 3 for all convex experiments. It accounts for up to 49% of the magnitude velocity at 75 kilometer depth in the model with the biggest amplitude ($A = 60$km) and $\beta = 1$. When $\beta = 2$ and with the maximum amplitude, the trench parallel velocity may even account for 98% of the total velocity field at a depth of 90 km (Fig. 5; Table 3). This trench parallel component of the velocity field is sufficient to allow transportation of heat and creates a symmetric pattern in the temperature field with a colder slab in the middle of the experiment. Our calculations show a difference of temperature of up to 110°C for the most extreme configuration tested (i.e. model SIN60_1 with $\theta = 36°$; Fig. 5a,c; Table 3).

**Variation of the curvature:** When varying the wavelength of the curvature of the experiments (e.g. set of experiments SIN40_2, SIN60_2, or -SIN40_2, -SIN60_2) the mantle flow pattern and thermal structure also show trench parallel variations. As in the reference model SIN20_1, the mantle flow is affected by the shape of the slab and shows some maximum trench velocity perturbation where the obliquity is maximum (e.g. $\theta = 17°$ at $x = 42$ km and $v_y = 12$ % of $\boldsymbol{v}$). This leads (1) to a difference in the velocity field, with $v_y$ representing up to 50% of the total velocity at 75 km depth and (2) to a variation of the plate interface temperature of about $\sim 140°$C (Table 3). This difference of temperature is observed in a distance of less that a hundred kilometers and is due to the strong trench parallel component of the mantle flow. Interestingly the coldest thermal regime in the convex models does not correspond to the edge of the modeling space where no trench parallel flow is allowed (see boundary condition on Fig. 2) but rather to the center. This is due to massive transport of mantle material towards the center of the convex slab.

**Concave models:** In the concave models, the mantle flow is directed towards the edges of the modeling space with a non-negligible trench parallel velocity. As a consequence, the coldest part of the model is located at the borders of the modeling space because mantle material is transported towards the model edges (Fig. 5b). The geometry of the modeling space is symmetric, similar to the reference model (See Fig. 5b). The temperature difference is the same as in the convex cases, and the velocity field show the same $v_y$ value at the same position (Table 3).

### 4.3.2 S-shaped models

The models described hereafter are named ATANA_$\gamma$ with reference to the parameters of Eq. 5. The maximum obliquity angle (see Eq. 5) is by definition located at the inflection point (i.e. at the center of the model). Its value evolves from $21°$ to $38°$ in our experiments. As seen in the previous section the shape of the box influences the pattern of the mantle corner flow. For all presented boxes, the mantle flow shows some deviation towards the right as depicted by the white arrows on the 3D view (Fig. 6: 3D views and $v_y$ maps) with a maximum trench parallel flow where the curvature is the most important. The trench-parallel velocity reach a maximum of 8.0 mm/yr in model ATAN40_05, of 1.8 mm/yr in model ATAN05_20 and of 3.5 mm/yr in model ATAN10_20. It corresponds respectively to $\sim 83\%$, $\sim 17\%$ and $\sim 35\%$ of the magnitude velocity at the same location. This differences in the velocity field lead to differences in the temperature field as represented on slices at 75 km depth and as depth-temperature path. Contrary to the convex and concave models, the temperature solution presented here is asymmetric as seen on 3 D view of the temperature field and on the the isotherm plotted on the slices at 75 kilometer depth (Fig. 6). It shows some important variation of the temperature along strike of the subduction zone too and this behavior is in agreement with the asymmetric mantle flow (Fig. 6).

We computed the maximum temperature difference between the center and side of the model to be of about $200°$C for models ATAN40_05 and ATAN10_20 ($\Delta T_{75km}$ equals $200°$C and $190°$ respectively; Fig. 6). In model ATAN40_05 the shape of the $450°$C isotherm is relatively smooth whereas in model ATAN10_20 it is sharper in direct relation with the shape of the trench. It shows that similar differences of temperature along strike can be obtained with different geometries. The model

ATAN05_20 has a maximum difference of 75°C between the middle and the coldest edge. The step in the shape of the 450°C isotherm is minimal. The comparison with model ATAN10_20 illustrates that increasing obliquity leads to increasing temperature variations.

## 5   Discussion

### 5.1   Implications of obliquity in subduction systems

We now evaluate the implications of our results for along-strike temperature variations in subduction zones with obliquity variations that consume a single plate. Our numerical experiments show a straightforward link between mantle wedge flow, the temperature at the plate interface, and the geometry of the subducting slab due to trench shape. This is observed for all type of geometries that we explored (convex, concave, S-shaped). The geometry affects the mantle wedge flow and adds a toroidal flow component to the dominant poloidal flow. This toroidal flow affects the temperature pattern at the plate interface. The temperature difference may become as much as ∼200°C in models with an obliquity of ∼40° (model ATAN40_05 or ATAN10_20; Fig. 6). These results agree well with previous numerical modeling work showing differences of temperature of ca. 100-200°C at 90 km depth (Bengtson and van Keken, 2012; Morishige and van Keken, 2014; Wada et al., 2015), ca. 120-350°C depending the depth (Ji and Yoshioka, 2015) or about 50°C at the base of the seismogenic zone (Yoshioka and Murakami, 2007). Our systematic study of the influence of the shape of the trench on the geotherm shows that a larger amplitude in the model (convex of concave) leads to a larger trench parallel flow and consequently a larger difference in the temperature at the plate interface. The S-shaped model is particularly interesting as it shows that even a small difference in geometry will be expressed as a trench parallel flow and a change of the temperature. The thermal regime is thought to be controlled by the angle of the subduction and the velocity and age of the downgoing plate, known as the $\Phi$ parameter ($\Phi = AV_n sin(\delta)$; with $A$ the age of the incoming lithosphere, $V_n$ the normal velocity of the incoming plate and $\delta$ the subduction angle; Kirby et al., 1991). Following our experiments (see Fig. 4), $\Phi$ remains certainly the first order parameter but we demonstrate that the trench parallel mantle flow influences on the temperature at the plate interface and may thus explain along-strike temperature differences in subduction zones. A 2D approach remains viable in systems with small obliquity, as stated in Bengtson and van Keken (2012), but important variations of geometry should be considered in further studies to reliably represent subduction zone dynamics both for present-day and past systems.

### 5.2   Limitations

The experiments with an amplitude variation of 20, 40 or 60 km over 250 km display a substantial effect on the mantle wedge flow and temperature pattern at the plate interface. Such geometrical variations are observed on Earth (e.g. in the Andean subduction around the border between Chile and Peru, or below Japan (Fig. 1)). However, our modelling approach is not without limitations: first, our model setting is a highly simplified geometry of a subduction zone; second, in our kinematic approach the deformation and thus shear heating (especially at the interface) is not taken in account; third, the isoviscous

rheology we use is known to underestimates the temperature predicted in the mantle wedge (van Keken et al., 2002). The hypothesis of isoviscosity also reduces the magnitude of the calculated trench parallel flow of at least one order of magnitude compared to a non-linear rheology for the mantle (Kneller and van Keken, 2008; Jadamec and Billen, 2012). This has naturally an effect on the temperature calculated in our models and allows us only to give only lower bounds estimates for the temperature variation at the plate interface. In addition no feedback mechanisms induced by effects of temperature change along the plate interface, such as water transport or melting processes that may influence the mechanical behavior, were taken in account. We made these simplifications because it allows for a dramatically lower computation time and was useful to evaluate the qualitative effect of the geometry on the thermal regime. We primarily aimed to test whether the major contemporaneous along-strike changes in temperature in geological records of subduction zones may be to first order explained by subduction obliquity changes, and our results suggest that they may indeed. Our study may thus form the basis for more detailed studies on the effect of obliquity on e.g. dehydration reaction and seismicity in subduction zones as function of obliquity, taking the effects of above limitations into account.

## 5.3    Comparison with the geological record

We now compare our models to the geologically constrained temperature variations in paleo-subduction zones. Our study was largely motivated by the geological record from Western and Central Turkey, but as mentioned, similar along-strike temperature variations have been recovered from other geological settings such as the Fanciscan complex (see review by Wakabayashi, 2015, 2017), the Sulawesi mélange (Parkinson, 1996), or the peri-Caribbean mélanges (Garcia-Casco et al., 2007). In all settings along-strike metamorphic grade variation were recorded at similar times within the same subduction system. Our model results suggest that variations in subduction obliquity, may tip a subduction interface from a (cold) gradient through the lawsonite blueschist facies to a (warm) gradient through the amphibole eclogite facies as calculated by Hacker et al. (2003a). Our models do not reproduce the extreme case of Anatolia, where the along-strike temperature variation at 30 km depth may have been as high as 500°C. However the reconstructed angle between the western and central Anatolian subduction segments may have been as much as 90°(Lefebvre et al., 2013; van Hinsbergen et al., 2016), and perhaps that extreme angle may explain the very high temperature gradient, although other factors, e.g. related to the continental nature of the downgoing plate, may have played a role. In addition, the geological record show that ridge spreading was present during the beginning of subduction in Central Anatolia (Maffione et al., 2017). This can account for part of the heat production in addition to the effect of the geometry. The limitation imposed by our ocean/ocean setup can also be taken in account. In the Turkish case there is a transition from an oceanic lithosphere subducting below an oceanic lithosphere to a continental lithosphere subducting below and oceanic one. This might affect the subduction dynamics if the continental ribbon is sufficiently large, e.g. > 200km (Tetreault and Buiter, 2012). This could be better tested using a dynamic model approach.

The influence of the mantle wedge convection pattern is interesting in terms of general understanding of subduction mechanism, their geochemical or geophysical structures and possibly their evolution. From a geological perspective it is interesting to reconcile field observation with models. Penniston-Dorland et al. (2015) argued that "rocks are hotter than models" but

since the shape of the subduction zone has an effect on the temperature at the plate interface (e.g Bengtson and van Keken, 2012; Morishige and van Keken, 2014; Ji and Yoshioka, 2015, this study) such differences may also be an artifact of the 2D geometry used in their study. Some models have shown that there is a strong competition between toroidal and poloidal flow around slab edges and that the velocity of the toroidal flow can be relatively large with respect to the poloidal flow (Jadamec and Billen, 2010; Király et al., 2017). Jadamec and Billen (2012) or Haynie and Jadamec (2017) even showed that the difference between the horizontal velocity field using linear and non-linear rheologies may be of more than one order of magnitude (6.5 mm/yr compared to 66 mm/yr for linear and non-linear respectively). Such changes in the velocity field would probably lead to differences in the temperature field itself and definitely increase the plate interface temperature. Modeling mantle flow accounting for differences in temperature is also of interest in the light of the *in-situ* record of temperature of the slab from melt-inclusion in arc eruptive volcanic rocks using geochemistry (Plank et al., 2009; Cooper et al., 2012). Such data suggest along-strike temperature variations may exist below Central America, Cascadia or the Marianas. Such data also question the temporal evolution of the thermal regime in subduction zones that was shown to evolve in time, for example in the Franciscan complex of California or in Western Turkey (Mulcahy et al., 2018; Pourteau et al., 2018) Finally, the obliquity effect is worth taking into account when assessing e.g. megathrust seismic hazards in link with dehydration reaction and events such as the episodic tremor and slip thought to represent fluid pulses along the interface in response to dehydration events (Rogers and Dragert, 2003; Audet and Kim, 2016).

In any case, the simple numerical modeling performed in this contribution positively tests our hypothesis that along-strike variation in subduction obliquity may have a first-order control on the temperature at the subduction interface, and may help us to understand the variation of geothermal gradient along strike of subduction zones, such as predicted by surface heat flow, for example below Japan (Tanaka, 2004). In light of the presented numerical models we argue that the plate boundary configuration may play a prime role in inducing strong lateral variations in geotherm within subduction systems, e.g. along the Aleutian trench, at kinks such as in Alaska or Kamchatka, and along the southern Marianas or northern Caribbean trenches.

## 6   Conclusions

Today's configuration of subduction zones, as well as plate tectonic reconstructions, show major along-strike variations in subduction obliquity along trenches. Here, we study the effect of trench geometry on temperatures at the subduction interface. To this end, we performed a series of simple numerical experiments with concave, convex and S-shaped subduction zones. Our results show that along-strike obliquity affects the geotherm of the subduction zone in two ways: by inducing a component of toroidal flow, and by changing the rate of subduction, both increasing temperatures at subduction zones with increasing obliquity. We compute along-strike temperature differences at 60-90 km depth that may be 200°C or more, depending on the geometry. This may be added to the well-known effects of subducting plate age to account for even larger temperature variations. On the other hand, our study did not take into account any feedback mechanisms induced by e.g. fluid flow or deformation, which may modify our estimates. In any case, we demonstrate that oblique trenches have the propensity to higher

geotherms. Our results may provide a basis to explain geological record of coeval metamorphic rocks that formed at the same subduction interface, but under very different pressure-temperature conditions (e.g. in Turkey, SE Asia, California). In addition, our study may be of importance for assessing the thermal regime of present-day subduction zones linked to melting processes and seismic hazards.

5   *Author contributions.* AP and DJJvH conceived the idea. AP and CT designed the experiments and AP carried them out. CT developed the finite element code ELEFANT. AP prepared the manuscript with contributions from all co-authors.

*Acknowledgements.* We thank Luuk Schuurmans for investigating several aspects of the presented research. Prof. Riad Hassani is warmly thanked for his help with the implementation of the stabilised $Q_1Q_1$ finite element formulation in ELEFANT. AP and DJJvH were funded through ERC starting grant SINK (306810). DJJvH acknowledges NWO Vidi grant 864.11.004. AP thanks Loes van Unnik Hoorn for 10   tremendous discussions during his postdoc at Utrecht University and the preparation of this manuscript. The perceptually-uniform color maps davos and vik were used in this study to prevent visual distortion of the data (http://www.fabiocrameri.ch/visualisation.php). Reviews by M. Faccenda and an anonymous reviewer were much appreciated and helped to strengthen the manuscript.

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

**Table 1.** Physical parameter used in the numerical model

| Symbol | Name | Value | Units |
|--------|------|-------|-------|
| $C_p$ | specific heat | 1250 | $J.kg^{-1}.K^{-1}$ |
| $\rho$ | volumetric mass density | 3300 | $kg.m^{-3}$ |
| $\mu$ | effective viscosity | $10^{22}$ | $Pa.s$ |
| $k$ | thermal conductivity | 2.5 | $W.m^{-1}.K^{-1}$ |

**Table 2.** Variation of temperature along strike for the reference model (SIN20_1) and those derived from it .

| Model Name Model Name | $v_{bc}$ $(mm/yr)$ | $\alpha$ | Slab age (My) | $\Delta T_{75km}$ °C |
|-----------------------|--------------------|----------|---------------|----------------------|
| SIN20_1 | 40 | 45 | 25 | 33 |
| SIN40_1 | 40 | 45 | 25 | 80 |
| SIN60_1 | 40 | 45 | 25 | 110 |
| SIN20_2 | 40 | 45 | 25 | 40 |
| SIN40_2 | 40 | 45 | 25 | 91 |
| SIN60_2 | 40 | 45 | 25 | 143 |
| 30dSIN20_1 | 40 | 30 | 25 | 50 |
| 30dSIN40_1 | 40 | 30 | 25 | 100 |
| 30dSIN60_1 | 40 | 30 | 25 | 160 |
| 18mm30dSIN20_1 | 18 | 30 | 25 | 50 |
| 44mm30dSIN20_1 | 44 | 30 | 25 | 50 |
| 56mm30dSIN20_1 | 56 | 30 | 25 | 50 |
| 50SIN20_1 | 40 | 45 | 50 | 33 |
| 75SIN20_1 | 40 | 45 | 75 | 33 |
| 100SIN20_1 | 40 | 45 | 100 | 33 |
| 70mmSIN20_1 | 70 | 45 | 25 | 31 |
| 20mmSIN20_1 | 20 | 45 | 25 | 34 |

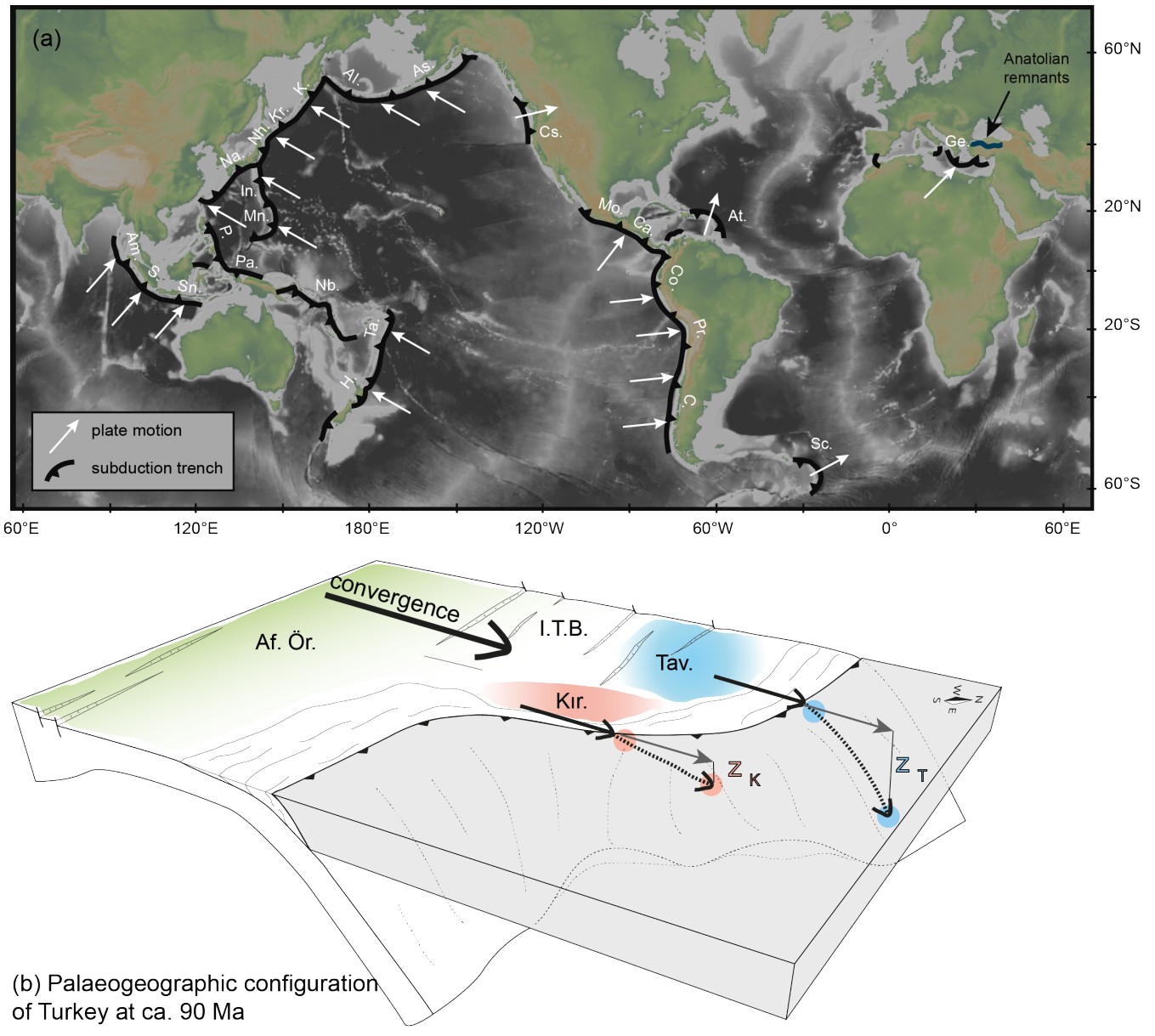

**Figure 1.** (a) Plate motion at trenches from the NNR-MORVEL model (Argus et al., 2011). Baselayer obtain with GeoMapApp (http://www.geomapapp.org) with topography and bathymetry from Ryan et al. (2009). Abbreviations as follow: Am., Andaman; As., Alaska; Al., Aleutians; At., Antilles; Ca., Central America; C., Chile; Co. Colombia; Cs., Cascadia; Ge., Greece; H., Hikurangi; In., Izu-Bonin; Nh., Japan; K., Kamchatka; Kr., Kurile; Mn., Marianas; Mo., Mexico; Nb., New Britain; Pa., Palau; Pr., Peru; P., Philippine; S. Sumatra; Sc. Scotia; Sn., Sunda; Ta., Tonga . (b) Possible palaeogeographic configuration at ca. 90 Ma for Central Turkey based on the reconstruction of van Hinsbergen et al. (2016). Abbreviation correspond to the following units: Af.Ör., Afyon-Ören zone; I.T.B. Inner Tauride Basin; Kır., Kırşehir block; Tav., Tavşanlı zone. $Z_K$ and $Z_T$ refer to the maximum burial of the Kirsehit and Tavşanlı units as discussed in text.

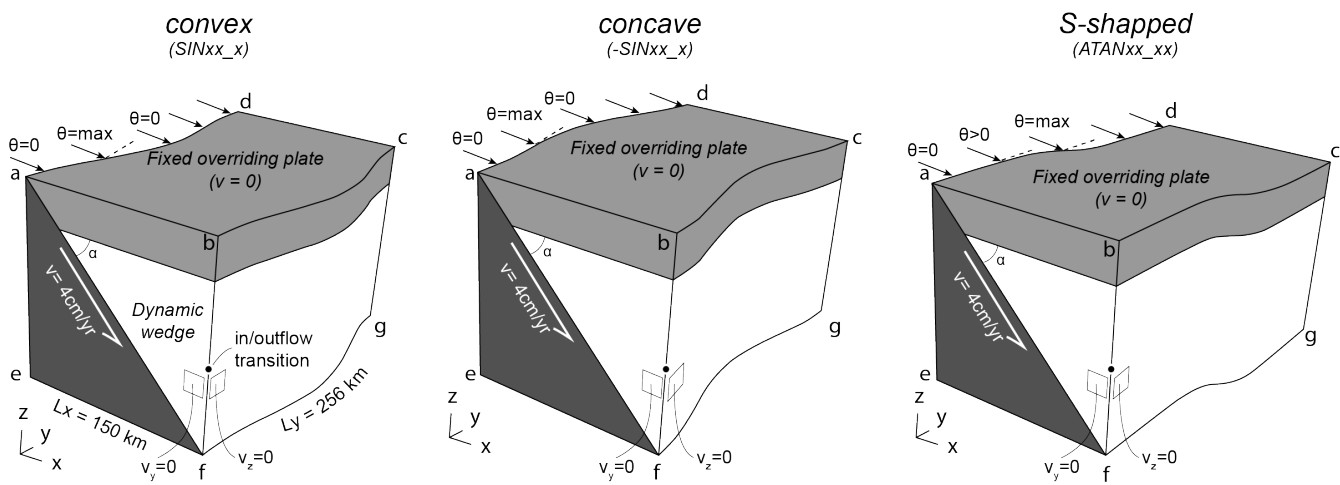

**Figure 2.** Setting of the computed models with the kinematic boundary condition. Initial thermal state is computed following the half-space cooling model following the formulation of Turcotte and Schubert (2014) with a 25 My old oceanic lithosphere for the downgoing slab and a 5 My old lithosphere for the upper plate. The physical dimensions are identical for each model and are specified on the convex setting. The number of elements is $65 \times 85 \times 65$ in the $x, y$ and $z$ direction respectively leading to a physical resolution of $2.3 \times 3 \times 2.30$ km for each models.

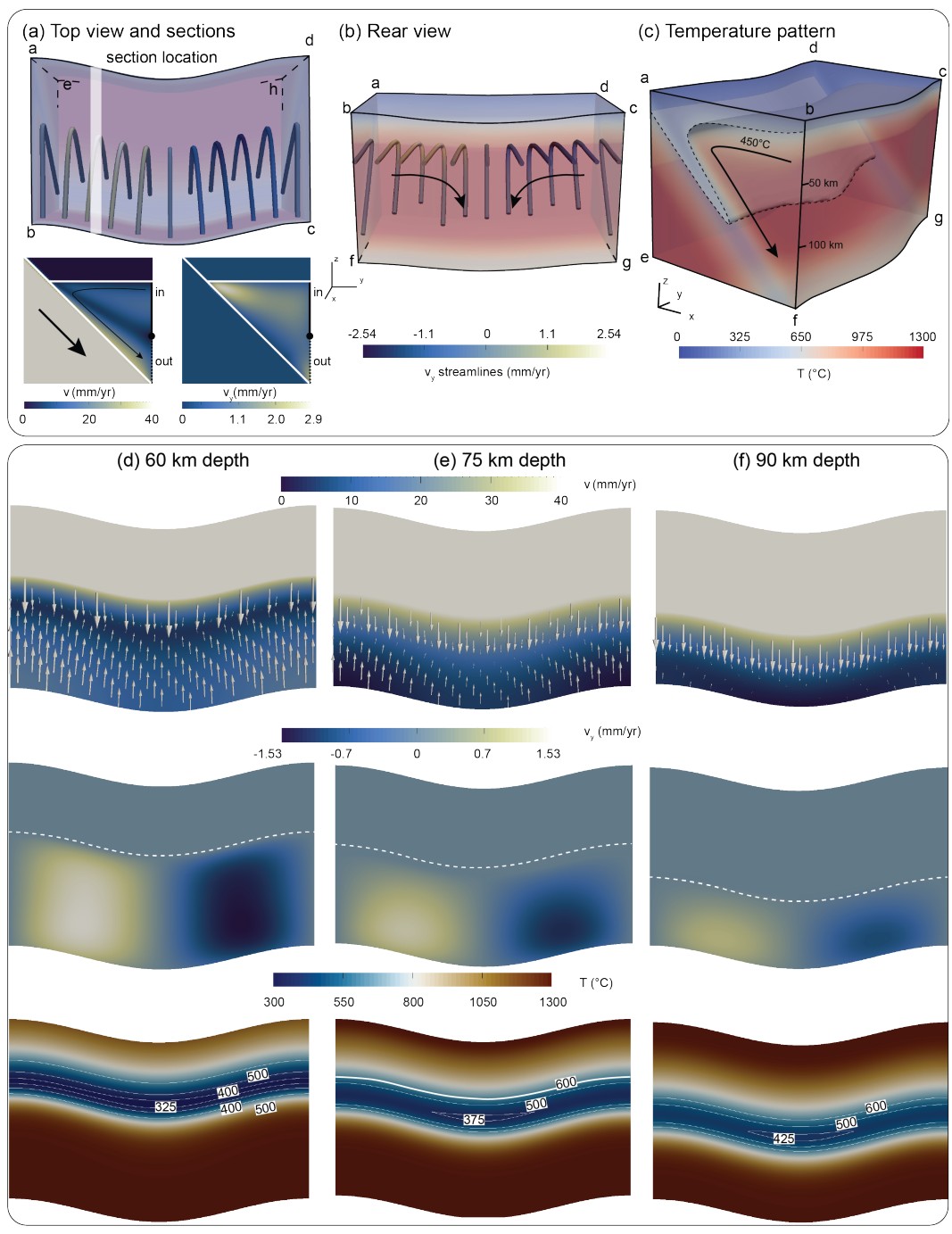

**Figure 3.** Results of the model SIN20_1. (a) Top view with streamlines showing the trench parallel mantle flow and sections $v$ and $v_y$ at $y = 64km$; (b) rear view of the domain with emphasis on the trench parallel flow represented as stramlines; (c) Temperature pattern in the model and deflection of the $450°C$ isotherm; (d) $v$, $v_y$ and $T$ at 60 km depth; (e and f) same as (d) at 75 and 90 km depth.

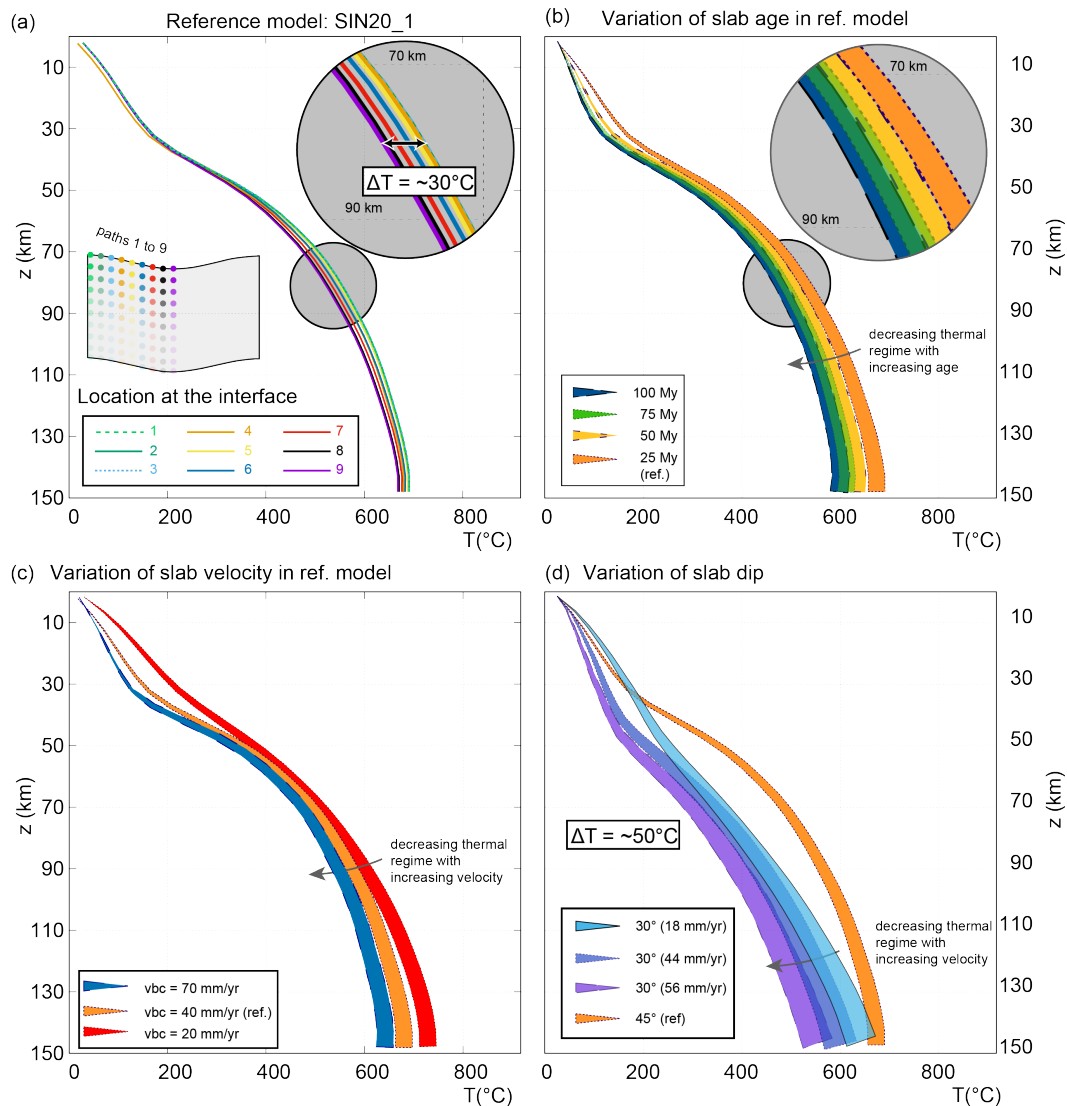

**Figure 4.** (a) Depth–Temperature path retrieved at the plate interface of the reference model (SIN20_1) with a downgoing plate velocity of 40 mm/yr. The relative position of the depth-temperature path is given (middle and edge). The zoom allows to better evaluating the relative position of the depth-temperature paths and the differences of thermal regime along-strike. The inset sketch gives the position of the sampled point along the slab with respect to depth (the color shading depict the deepening); (b) Variation of thermal regime with respect to slab age. The blue, green, yellow and orange curves give the temperature range in the model (ca. 30°C); (c) investigation of different subducting rates for the reference experiment; blue is 70 mm/yr and red is 20 mm/yr; (d) variation of temperature for a subduction angle of 30°investigated for different subduction rates.

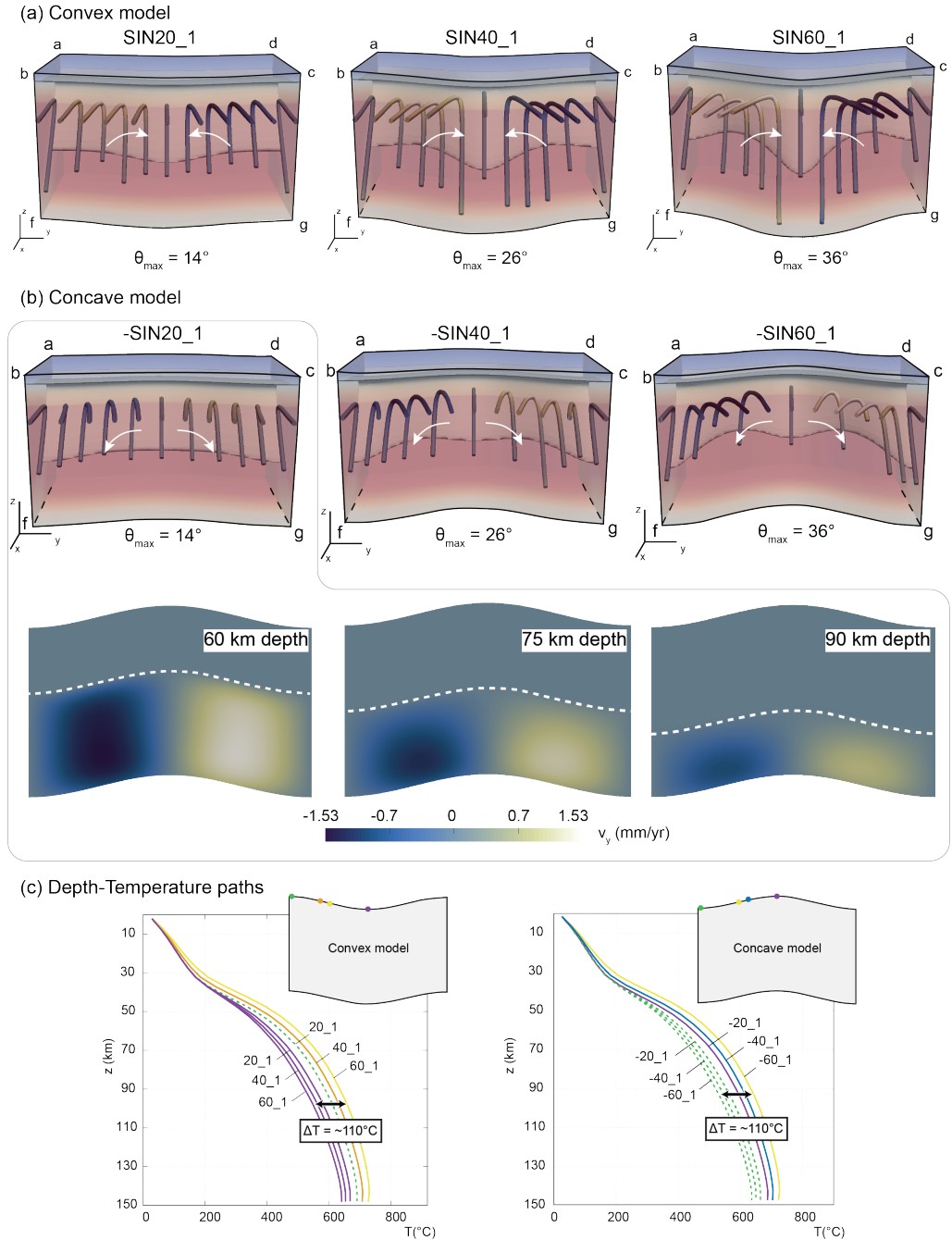

**Figure 5.** (a) Mantle flow, shape of the $450°$C isotherm and depth temperature path for the convex models (SIN20_1, SIN40_1 and SIN60_1); (b) Mantle flow, shape of the $450°$C isotherm and depth temperature path for the concave models (SIN20_1, SIN40_1 and SIN60_1). Velocity map of the $y$ component are reported at 60, 75 and 90 km depths for the case -sSIN20_1. (c) summary of the depth-temperature path calculated for the convex and concave models, showing a maximum temperature variation of ca. $110°$C with the most oblique models. Details of the $\Delta T_{75km}$ are given in Table 2.

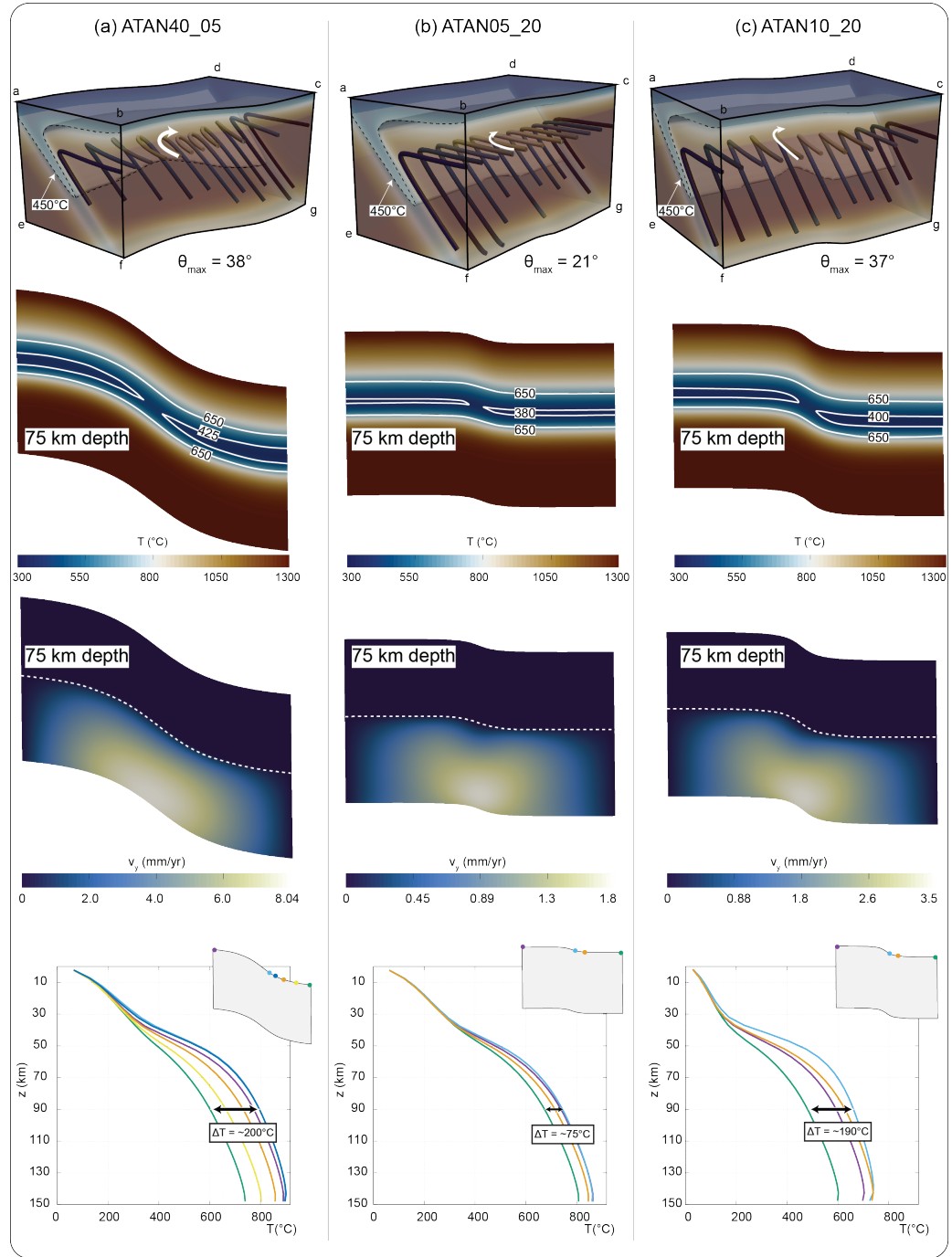

**Figure 6.** From to to bottom: 3D view of model ATANA_$\gamma\gamma$ showing the mantle flow streamlines and the contour of the $450°$C isotherm and its more or less important deflection at the center of the modeling space. The white arrows emphasize the direction of the mantle flow. They are dimensionless. Map of temperature at 75 km with isotherms in white. Map of trench parallel velocity at 75 km depth. Depth–temperature path along the plate interface showing a $\Delta T$ of 200 °C, 75 °C or 190 °C depending on the model. (a) Model ATAN40_05; (b) Model ATAN05_20; (c) Model ATAN10_20

**Table 3.** Variation of $v_y$ in the mantle at different depth compared with the magnitude velocity at the same position. The variation of temperature at 75 km depth is also give to complete table 2

| Model name | $\Delta T_{75km}$ °C | depth km | $v_y$ | $v$ mm/yr | $v_y/v$ |
|---|---|---|---|---|---|
| SIN20_1 | | 60 | 1.54 | 13.2 | 12% |
| $\theta_{max}$ 14° | 30 | 75 | 1.42 | 8.77 | 16% |
| $y_{\theta_{max}}$ ±64 | | 90 | 0.944 | 4.54 | 17% |
| SIN20_2 | | 60 | 1.51 | 15.0 | 10% |
| $\theta_{max}$ 17° | 40 | 75 | 1.28 | 10.5 | 12% |
| $y_{\theta_{max}}$ ±42 | | 90 | 0.976 | 4.13 | 24% |
| SIN40_1 | | 60 | 3.08 | 10.4 | 23% |
| $\theta_{max}$ 26° | 80 | 75 | 2.54 | 8.30 | 30% |
| $y_{\theta_{max}}$ ±64 | | 90 | 1.88 | 4.73 | 39% |
| SIN40_2 | | 60 | 3.26 | 15.9 | 21% |
| $\theta_{max}$ 32° | 91 | 75 | 2.76 | 11.0 | 25% |
| $y_{\theta_{max}}$ ±42 | | 90 | 2.14 | 3.79 | 56% |
| SIN60_1 | | 60 | 4.68 | 12.3 | 38% |
| $\theta_{max}$ 36° | 110 | 75 | 3.81 | 7.76 | 49% |
| $y_{\theta_{max}}$ ±64 | | 90 | 2.84 | 4.78 | 59% |
| SIN60_2 | | 60 | 5.37 | 16.2 | 33% |
| $\theta_{max}$ 43° | 143 | 75 | 4.58 | 10.9 | 42% |
| $y_{\theta_{max}}$ ±42 | | 90 | 3.59 | 3.67 | 98% |
| ATAN40_05 | | 60 | 8.65 | 10.3 | 84% |
| $\theta_{max}$ 38° | 200 | 75 | 8.04 | 9.68 | 83% |
| $y_{\theta_{max}}$ 128 | | 90 | 6.54 | 6.87 | 95% |
| ATAN05_20 | | 60 | 1.88 | 13.0 | 14% |
| $\theta_{max}$ 21° | 75 | 75 | 1.78 | 10.0 | 17% |
| $y_{\theta_{max}}$ 128 | | 90 | 1.49 | 4.01 | 37% |
| ATAN10_20 | | 60 | 3.74 | 13.8 | 27% |
| $\theta_{max}$ 27° | 190 | 75 | 3.50 | 10.0 | 35% |
| $y_{\theta_{max}}$ 128 | | 90 | 2.98 | 3.79 | 79% |