# Peer review of "The effect of obliquity on temperature in subduction zones: insights from 3D numerical modeling."

_Solid Earth, 2017_

## Referee Comment (RC1) · M. Faccenda (Referee) · 3 Feb 2018

Plunder and co-authors have addressed the role of subduction obliquity in modifying the slab thermal structure. They found that trench-parallel (toroidal) component of the subduction-induced mantle flow can generate from  $50^{\circ}$  to  $200^{\circ}$  along-strike temperature differences according to the subduction velocity, and, more importantly, the subduction obliquity.

The manuscript represent a substantial contribution to scientific progress within the scope of Solid Earth, as it suggests that along-strike variations in the degree of metamorphism in exhumed rocks (assuming that the pressure represent depth and no important contribution derives from tectonic stresses) could be explained by subduction

obliquity.

The scientific approach and applied methods are valid, and the model limitations are fairly discussed. Results are concise and clearly explained.

The only major comment I have is that, in order to make the paper more appealing to a wider geological and geophysical audience, may be the authors could have investigated how the results change as a function of the (i) slab dip (for example, 30-60-90 degrees), (ii) slab age (for example, 50-75-100 Myr) and (iii) upper plate age (5 Myr is a quite unusual age for the upper plate where oceanic plates subduct below overriding continents). In this way the results could be more applicable to different subduction settings, and successively could be further tested in another study by introducing further complications like dehydration and melting reactions, temperature- and compositiondependent viscosity, etc.

Aside this, I recommend publication of the paper in the present form.

ć Page 7, line 10: typos. ć Page, line 25: vy/v = 2.55/30 = 8.47%. Why 2.9%? ć Page 10, line 26: typo

Manuele Faccenda

---

## Referee Comment (RC2) · Anonymous Referee #2 · 26 Mar 2018

Review:

The study presented here investigates possible temperature variations at the subduction interface due the subduction obliquity. The motivation comes from geological data (Western and Central Turkey) and present-day configuration of the global subduction system, that large slab segments are subducting at an angle relative to their upper plate. The authors perform 3-D thermo-kinematic numerical models, in which they vary: a) the curvature (convex, concave, sinusoidal), b) amplitude of curvature (sinA), c) and parameters beta and gamma, that control the shape of the curvature. Two additional simulations were performed on a subset (reference model, convex geometry) to investigate the effect of the subduction rate.

The focus of the study is thus calculating the flow in the mantle wedge and the temperature variations at the subduction interface (how temperature profiles vary laterally). Results show that the effect of the trench curvature (obliquity) on the geotherm is considerable. Variations in obliquity can lead to temperature variations as large as 200degC along strike. The results are then discussed in relation to geological data from the Western and Central Turkey, and could potentially be applied to other presentday/paleo-oblique subduction zones.

The manuscript at this point has a well defined structure, with clear and well documented results and conclusions. Some exceptions include insufficient figure captions, labels that need to be improved, and few paragraphs that need rephrasing/more details.

I recommend the manuscript to be published in Solid Earth with major modifications, and I identify below 5 major points to be addressed, followed by other minor points. My comments primarily aim at clarifying some aspects of the model and results, and thus making the manuscript a more complete piece of work.

Major points:

1. Model details.

a) Time stepping and temperature advection. Temperature advection (i.e. Page 8, Line 5) was suggested in a couple of locations as an important mechanism. However, it is not explained what temperature advection is (for the general audience), or how you solve for it (from Eq. 3).

A few questions to help here: When was steady-state (Abstract, Line 8) reached in simulations? How long did the models run? Did you solve just once for Stokes and T equation (1 time step)? How large was the time step? This is not clear. What about transient evolution of temperature and feedback to the system (i.e. flow of hot material that facilitates subduction)?

b) Model dimensions. What are the physical dimensions of the model? What is the physical resolution of the domain? Box dimensions are indicated in Fig 2a, but please

SED
include more information in the main text.

c) Inflow/outflow boundary conditions (i.e. Page 6, Line 10). Could you explain better the in/out flow condition at 100 km depth? Did you choose this particular BC to allow for corner flow in the mantle wedge? If so, please indicate in text.

2. Subduction curvature and obliquity. Confusing interchange of "curvature" and "obliquity". For example, Page 1, Line 10: One sentence uses "trench curvature", and the next "obliquity". Authors should make it clearer how obliquity and curvature are linked to each other.

Global subduction zone. Page 2, Line 32, Figure 1: Measured subduction curvature depends on the trench length considered (i.e. Schellart et al 2007). When you make the statement "majority of subduction zones have concave,... or convex", do you consider the length? What is the maximum curvature/obliquity (i.e. theta\_max) for the present-day natural system?

3. Systematic study. First, all simulations (convex, concave, s-shaped) should be clearly listed as in Table 2, with corresponding varied parameters. Then, comparing the model results as in Table 2 across the entire simulation spectrum (v, vy\_max, dT\_max) could provide more information on the general behaviour of the system. For example, that the largest dT are obtained for s-shaped simulations, what are the max/min bounds for dT for each geometry, or how subduction rate affects dT. I consider valuable information could be derived from an extended Table 2.

4. The study needs more link to former studies on the topic. For example, Page 2, Line 16, Lines 24-25: Ji and Yoshioka (2015), Yoshioka and Murakami (2007) also investigated the relationship between slab geometry (convex, concave) and obliquity, and the thermal regime of the plate interface in 3D models. It deserves more explanations (what did they find, what is different in your model etc.) than just a mention. A comparison between their and your results should also be included in the manuscript.
5. The limitations of the model need to be discussed into more details (i.e. Page 10, Lines 24-27). What are the factors that could modify your results (max dT  $\sim$ 200C)? i.e. revision of field data (different P-depth interpretation), model improvements (non-linear rheology for mantle), geometry, subduction parameters (age, velocity, subduction angle) etc. What about dynamic models (steady-state vs transient state of temperature)?

Minor points:

Page 1.

Line 1: The geotherm in subduction zones. Line 4: proposed/observed instead of supposed. Line 7: Please revise the sentence: some commas missing and remove "only". Line 8: the results in terms of: (i) mantle flow..., and (ii) temperature... Line 12: heat that is advected by velocity causes such temperature variations (linked to the magnitude of the trench parallel component of velocity). Line 17: are primarily. Lines 19-23: Sentence too long, please rephrase. Line 24: "with" instead of "whereby".

Page 2.

Line 2: trench perpendicular flow (poloidal). Line 11 (paragraph): Explain what is temperature advection and why it is important/of interest here?. Line 13: trench curvature vs obliquity - should be explained what they are/stay consistent. Line 22: setup.

Page 3.

Line 6: with increasing. Line 10: proxies to record them [lateral variations in temperature]. Line 14: Melt-inclusion data suggest that temperature variations occur along strike and vary through <time>. This invites for some discussion about time-dependent (dynamic) variations in slab geometry. Line 16-18: Please explain what's the difference between eclogite and garnet-amphibolite facies (i.e. high/low P,T) for the general audience. Line 18: Yamato and Brun (2016) have shown that peak pressures recorded in subducted rocks might not reflect their maximum burial depths. This suggests that the assumption of transforming pressure into depth might not be the best practice. Could SED
you comment on this aspect? How would that change the temperature variation estimated in line 19 (i.e. >300C) and how would that relate to your modeling results? Line 27: Please explain in a few words what is supra-subduction, as compared to subduction for the general audience.

**Page 4.**

Lines 3-7: Please rephrase this sentence. It is too long. Line 10: Nice transition/motivation to the next section. Line 15: measurements. Paragraph 22-33: This paragraph provides some background on previous studies investigating the effect of geometry (obliquity) on subduction dynamics. However, more should be included on studies that look at development of trench curvature (i.e. Schellart et al - convex/concave due to slab width, or sinusoidal when there is both trench advance and retreat), because these studies are more relevant to the present investigation.

**Page 5.**

Line 8: Why no analytical solution in 3D? Perhaps because of its too complex nature (i.e. take into account poloidal and toroidal components and other complex features)? Also, a lot of work was restricted to 2D in previous decades because of computer limitations at that time. In principle, 2D is a first order approximation, which yielded some important results, but with some limitations. The authors could explain why the transition from 2D to 3D studies (i.e. trench obliquity is an inherent 3D feature). Line 9: setup. Line 23: parameter values ...Table 1. Line 24: Decoupled energy equation: what about dislocation+diffusion creep, with P,T dependence for mantle viscosity? How would that affect temperature advection on the interface? Line 32 (throughout manuscript): Need to be consistent with units, especially for the time unit (yr): Ma, My, Myr, cm/yr etc. Model Setup: Should indicate before line 30, that that the model setup and boundary conditions are tuned for the Anatolian case study explained in Section 2.

Page 6.
Line 7: than. Line 16: use. Line 15-19: Any computational libraries that need to be cited here? Line 20: How deformed are the Q1Q1 elements to conform with the geometry? Is that affecting the accuracy of the solution?

Page 7.

Line 9: multiple typos. Why use these particular boundary conditions? Is mass conserved? The inflow/outflow bc are not clearly explained (i.e. flow comes in horizontally from the top right boundary and flows out at the bottom boundary, conserving the mass... Line 24: sentence is not clear. Line 28, Figure 3: Why does the magnitude of vy\_max decrease with depth? Is it a consequence of the model setup?

Page 8.

Line 4 (end): Reference to Figure 4 is incorrect, as that observation was derived from Figure 3. Line 6: becomes. Figure 4: insufficiently explained in the text/caption. What are the pink/light blue lines in figure 4a? It also seems in Figure 4b, that path 3 is the warmest (compared to 1,2,4) - which corresponds to location theta\_max? Line 14: What was the subduction rate for the reference model (sin20\_1) compared to these values? Line 15-16: Why an increase in subduction velocity produces such a polarity change in the temperature variation? This is not explained why that happens/results are not shown. The paragraph is not clear that it refers to the pink/light blue lines in Fig 4. Also, please clarify the differences between the reference model and these additional experiments. Line 20: also called inflection points.

Page 9.

Line 3: 100 or 110C (as in the Fig5c)? Line 6: centre (purple contours), flow brings colder material from the surface to the centre of the slab. Concave models: I feel their model description is incomplete. The warmest part of the slab is in the center. How warm relative to the edges? What about vy\_max? Line 16: max theta is in the inflection point. Line 20: sentence not clear. Lines 21-22: typos. Line 23: Why is the

SED
temperature field asymmetric in Figure 6b, slice at 75 km depth, as compared to a,c? Line 26, Figure 6: T variation for ATAN05\_20 (fig 6b) is indicated 75C, not 200C as suggested in the text./ If you mean ATAN10\_20, the T variation in fig 6c is indicated 110C, even if it looks around 200C. Please revise figure 6 and paragraph.

**Page 10.**

Line 9: In which way the results in this paper agree well with previous work? Line 16: Theta remains constant in the simulations here. Should make it clear and potentially discuss implications for variable Theta (age, velocity, subduction angle) Lines 24-27: The limitations should be extended a bit more. For example, how would power-law T,p dependent rheology of the mantle expect to influence the T variations? Are the T variations calculated here lower bound estimates? Line 25: typo.

Page 11.

Lines 11-12: What are the limitations/improvements of the numerical model that could produce a larger T variation as observed in Anatolia?

Figures and Tables.

Figure 1: Please add some labels/names of the major subduction zones used to illustrate in the text (i.e. Aleutians, remnants of W-C Turkey). b) The name abbreviations in the figure are not clear. Please either explain in caption or in figure.

Figure 3: Labels for Fig 3g,h,i are missing (as indicated in the text on Page 8, line 2)

Figure 4: Incomplete figure caption/figure legend. Labels are missing (a,b). What do the paths,vbc,"middle/edge" represent? Some sentences in the caption could help explain the figure better.

Figure 5: Please indicate simulation labels (i.e. sin20\_1) in Fig 5a-b. c) The colours for the maximum temperature paths are misleading. If they are all taken at the inflection point (yellow in the sketch), they should all be yellow (like the purple curves). Similarly
for the concave model.

Figure 6: Vy direction arrows as in Figure 5a,b would be useful.

---

## Author Response (AR1)

**Answers to the interactive review of the paper: "The effect of obliquity on temperature in subduction zones: insights from 3D numerical modeling" by Alexis Plunder et al.**

**Dear Editor, dear reviewers,**

First of all, we wish to thanks the reviewer for their time considering our work. Below you can find point by point *answers* the to questions asked by the referees.

Below can be found a marked version of the manuscript with all changes made

**Answers to first referee: M. Faccenda (Referee) manuele.faccenda@unipd.it**

Received and published: 3 February 2018

Plunder and co-authors have addressed the role of subduction obliquity in modifying the slab thermal structure. They found that trench-parallel (toroidal) component of the subduction-induced mantle flow can generate from 50 to 200 along-strike temperature differences according to the subduction velocity, and, more importantly, the subduction obliquity. The manuscript represent a substantial contribution to scientific progress within the scope of Solid Earth, as it suggests that along=strike variations in the degree of metamorphism in exhumed rocks (assuming that the pressure represent depth and no important contribution derives from tectonic stresses) could be explained by subduction obliquity.

The scientific approach and applied methods are valid, and the model limitations are fairly discussed. Results are concise and clearly explained. The only major comment I have is that, in order to make the paper more appealing to a wider geological and geophysical audience, may be the authors could have investi- gated how the results change as a function of the (i) slab dip (for example, 30-60-90 degrees), (ii) slab age (for example, 50-75-100 Myr) and (iii) upper plate age (5 Myr is a quite unusual age for the upper plate where oceanic plates subduct below overriding continents). In this way the results could be more applicable to different subduction settings, and successively could be further tested in another study by introducing further complications like dehydration and melting reactions, temperature- and composition- dependent viscosity, etc.  $\backslash$

Concerning (i): we now provide a set of experiment for additional subduction dip angle (30°). As expected, the results show similar patterns to what is described for the experiments presented in the original manuscript (with 45°). The corresponding depth-temperature paths were added on Fig. 4. We excluded subduction dip angles of 60° and 90° for problems related to the setup and their respective boundary conditions: With a dip angle of 60° we have seen large effects on the side of the

models and therefore excluded them. A dip angle of 90 ° is so rare that it is beyond the scope of our investigation.

(ii) We now provide the depth-temperature path for the reference model with different slab ages (Fig. 4). As expected, older lithosphere decreases the thermal regime of the subduction zone but does not change the lateral effect that is the prime target of our study. We want to warn the reviewer that the one-to-one applicability of our model to natural settings would be difficult considering the large assumptions we made. Rather, we perform a set of experiments to test whether the hypothesis that strong lateral changes in metamorphic grade along a subduction system may reflect changes in obliquity, is physically plausible. Our experiments suggest that it is.

(iii) We chose the young overriding plate age in order to be as close as possible to the geological observation of Turkey (ophiolite forming during subduction initiation). This is now better stated. It also allows us to reduce the size of the computational domain (and the size where v=0).

The geometries used in our paper are suit the testing of our hypothesis, but should be adjusted if designed for direct comparison to the field in future work.

Aside this, I recommend publication of the paper in the present form.

The typos have been found, corrected. We also double-checked carefully the manuscript.

We removed one line of the comments by ref. \#1 due to LaTeX incompatibility

Manuele Faccenda

**Answers to second referee**

Anonymous Referee \#2, Received and published: 26 March 2018.

Review: \\

The study presented here investigates possible temperature variations at the subduction interface due the subduction obliquity. The motivation comes from geological data (Western and Central Turkey) and present-day configuration of the global subduction system, that large slab segments are subducting at an angle relative to their upper plate. The authors perform 3-D thermo-kinematic numerical

models, in which they vary: a) the curvature (convex, concave, sinusoidal), b) amplitude of curvature (sinA), c) and parameters beta and gamma, that control the shape of the curvature. Two additional simulations were performed on a subset (reference model, convex geometry) to investigate the effect of the subduction rate. The focus of the study is thus calculating the flow in the mantle wedge and the temperature variations at the subduction interface (how temperature profiles vary later- ally). Results show that the effect of the trench curvature (obliquity) on the geotherm is considerable. Variations in obliquity can lead to temperature variations as large as 200degC along strike. The results are then discussed in relation to geological data from the Western and Central Turkey, and could potentially be applied to other present- day/paleo-oblique subduction zones. The manuscript at this point has a well defined structure, with clear and well docu- mented results and conclusions. Some exceptions include insufficient figure captions, labels that need to be improved, and few paragraphs that need rephrasing/more details. I recommend the manuscript to be published in Solid Earth with major modifications, and I identify below 5 major points to be addressed, followed by other minor points. My comments primarily aim at clarifying some aspects of the model and results, and thus making the manuscript a more complete piece of work. Before answering in details, we note that the revised version of the manuscript has been carefully checked for possible remaining typos and mistakes. The figures have been revised considering the comments of referee |#2, and captions have been re-written in more detail.

Major points: 1. Model details. a) Time stepping and temperature advection. Temperature advection (i.e. Page 8, Line 5) was suggested in a couple of locations as an important mechanism. However, it is not explained what temperature advection is (for the general audience), or how you solve for it (from Eq. 3).

Temperature advection, of more generally advection is the mechanism of transporting a quantity (vectorial or scalar). It is explained by the equation (\#3) itself.

 $v.\underline{nabla} = \underline{vx} d/\underline{dx} + \underline{vy} d/\underline{dy} + \underline{vz} d/\underline{dz}$ where v is the velocity field.
Then the temperature (scalar) advection becomes the following vector:  $(v \cdot \underline{nabla})T = [\underline{vx} dT/\underline{dx} + \underline{vy} dT/\underline{dy} + \underline{vz} dT/\underline{dz} \\
 \underline{vx} dT/\underline{dx} + \underline{vy} dT/\underline{dy} + \underline{vz} dT/\underline{dz} \\
 \underline{vx} dT/\underline{dx} + \underline{vy} dT/\underline{dy} + \underline{vz} dT/\underline{dz} ]$

We are not sure whether advection really needs to be explained in a research paper. It has been explained in many textbooks and is rather a simple notion. In any case, all methods are explained in the appendix of the paper by co-author Thieulot (Thieulot 2011, PEPI)

A few questions to help here: When was steady-state (Abstract, Line 8) reached in simulations? How long did the models run? Did you solve just once for Stokes and T equation (1 time step)? How large was the time step? This is not clear. What about transient evolution of temperature and feedback to the system (i.e. flow of hot material that facilitates subduction)?

The Stokes equation is solved once (because we use a linear fluid). Concerning the steady state, we use a similar approach as mention in Currie et al. 2004, Wada & Wang 2009, of Kneller & van Keken 2008. This is now better stated in the manuscript. The time step is changing to respect a CFL condition (order of 5000 yr). The steady state is reached after 15-20 My, depending on the initial geometry. This is now stated in the manuscript. Because the fluid is linear viscous there is no feedback in the system.

The simulations are run until the temperature pattern in the slab is not mainly driven by the advection term of the energy equation (eq 3). We run the calculations until steady state is reached (ca. 15-20 Ma depending on the simulation with a time step of about 5000 years) on a Desktop machine using a single processor. Each model took about one to two hours to compute.

b) Model dimensions. What are the physical dimensions of the model? What is the physical resolution of the domain? Box dimensions are indicated in Fig 2a, but please include more information in the main text.  $\backslash$

The physical dimensions of the models are provided on Fig. 2. They are now included in the manuscript. The physical resolution is  $\sim 2.30-3-2.30$  km and is now included in the model setup description.

c) Inflow/outflow boundary conditions (i.e. Page 6, Line 10). Could you explain better the in/out flow condition at 100 km depth? Did you choose this particular BC to allow for corner flow in the mantle wedge? If so, please indicate in text. \\

This is a "classical" approach for such models. As in other studies (van Keken et al., 2002; Curie et al. 2004; Wada et Wang 2009, or Wada et al., 2015) we prescribe an in/out flow boundary condition to allow for corner flow. This is now stated in the manuscript.

2. Subduction curvature and obliquity. Confusing interchange of "curvature" and "obliq-uity". For example, Page 1, Line 10: One sentence uses "trench curvature", and the next "obliquity". Authors should make it clearer how obliquity and curvature are linked to each other. \\

**True.**

We now clarified: "Real subduction zones, however, tend be curved, i.e. trench strike varies laterally and the angle between the absolute plate motion at the trench and trench strike – the subduction obliquity – thus change along-strike".

Global subduction zone. Page 2, Line 32, Figure 1: Measured subduction curvature depends on the trench length considered (i.e.

Schellart et al 2007). When you make the statement "majority of subduction zones have concave,. . or convex", do you consider the length? What is the maximum curvature/obliquity (i.e. theta max) for the present-day natural system?  $\$

Thanks for raising that question. We just start from an easy observation. Trenches of subduction zones have shapes, and the majority of these shapes are concave or convex. The theta max today can be \$>45^{\circ}\$ (In the Marianas for example). There are even extreme cases such as the Northen part of the Sunda-Sumatra system or the Aleutian trench, where the plate boundary becomes a transform fault.

The additional text reads: Some trenches contains as much as 90\$^{\circ}\$ curvature such that along the same trench, subduction may gradually (Aleutians, Sunda-Burma) or abruptly (southern Marianas, northern Lesser Antilles) change from near-orthogonal subduction to near-transform motion.

3. Systematic study. First, all simulations (convex, concave, sshaped) should be clearly listed as in Table 2, with corresponding varied parameters. Then, comparing the model results as in Table 2 across the entire simulation spectrum (v, vy max, dT max) could provide more information on the general behaviour of the system. For example, that the largest dT are obtained for s-shaped simulations, what are the max/min bounds for dT for each geometry, or how subduction rate affects dT. I consider valuable information could be derived from an extended Table 2.  $\backslash$

Table 2 was replaced by table 2 and 3 and was completed. We also added / completed figure 4 with depth-temperature paths of other experiments. Table 2 shown the variation of the reference model with respect to: subduction angle, age of the downgoing plate, velocity of the downgoing plate, and variation of geometry of the ref setup. Table 3 completes the former table 2 with all the settings.

4. The study needs more link to former studies on the topic. For example, Page 2, Line 16, Lines 24-25: Ji and Yoshioka (2015), Yoshioka and Murakami (2007) also investigated the relationship between slab geometry (convex, concave) and obliquity, and the thermal regime of the plate interface in 3D models. It deserves more explana- tions (what did they find, what is different in your model etc.) than just a mention. A comparison between their and your results should also be included in the manuscript. \\

A more systematic discussion with respect to previous models is now provided. The reference to Ji and Yoshika (2015) was better made. Concerning the other reference, we thank the reviewer as we have missed it. It is now added in the manuscript and discussed: These results agree well with previous numerical modeling work showing differences of temperature of ca. 100-200 deg. at 90 km depth (Bengtson2012,

Morishige2014, Wada2015), ca. 120-350 depending the depth (Ji2015) or about 50 deg. C at the base of the seismogenic zone (Yoshioka2007).

5. The limitations of the model need to be discussed into more details (i.e. Page 10, Lines 24-27). What are the factors that could modify your results (max dT 200C)? i.e. revision of field data (different P-depth interpretation), model improvements (non-linear rheology for mantle), geometry, subduction parameters (age, velocity, subduction an- gle) etc. What about dynamic models (steady-state vs transient state of temperature)? \\

The fieldwork data for the Turkish case are pretty rock solid (different study by different groups over the last decades). So we are not sure the model (as simplified as they are) can really question them. The non-linear rheology would definitely be a nice addition to our modeling setup. Having a better geometry (a real subduction zone geometry) would also be a plus. Dynamic models would also probably lead to along-strike temperature differences. This could be a next step for our study building on the work of many authors by adding temperature in their dynamic models of subduction zone.

That said: the aim of our study is a test of physical plausibility, or, in other words, an attempt to falsify the hypothesis that subduction obliquity caused lateral temperature variation. We did not falsify that hypothesis, pending the assumptions we made. Future work may succeed in falsifying the hypothesis if more details are taken into account, but for now, our hypothesis stands.

Minor points: Page 1. \\ Line 1: The geotherm in subduction zones. Changed Line 4: proposed/observed instead of supposed. Replaced with "proposed" Line 7: Please revise the sentence: some commas missing and remove "only". done

Line 12: heat that is advected by velocity causes such temperature variations (linked to the magnitude of the trench parallel component of velocity).

**Sentence changed**

Line 17: are primarily. Lines 19-23: Sentence too long, please rephrase. Line 24: "with" instead of "whereby".

The sentence has been shorten and revised

Page 2.
Line 2: trench perpendicular flow (poloidal).
Changed

Line 11 (paragraph): Explain what is tem- perature advection and why it is important/of interest here? We have added a sentence

Line 13: trench curvature  $\underline{\rm vs}$  obliquity – should be explained what they are/stay consistent.

See earlier comment: we have explained this. In our view, the difference is pretty obvious: curvature is the along-strike change in trench strike, obliquity is the angle between absolute plate motion of the downgoing plate and the trench

Line 22: setup. Corrected

Page 3. Line 6: with increasing. Corrected

Line 10: proxies to record them [lateral variations in temperature]. *Corrected*

Line 14: Melt-inclusion data suggest that temperature variations occur along strike and vary through <time>. This invites for some discussion about time-dependent (dynamic) variations in slab geometry.

This is now stated in our discussion

Line 16-18: Please explain  $\frac{what's}{(i.e. high/low P,T)}$  for the general audience.

A brief notice is now provided.

Line 18: Yamato and Brun (2016) have shown that peak pressures recorded in subducted rocks might not reflect their maximum burial depths. This suggests that the assumption of transforming pressure into depth might not be the best practice. Could you comment on this aspect? How would that change the temperature variation estimated in line 19 (i.e. >300C) and how would that relate to your modeling results? \\

The point raised by Yamato and Brun (2016) is really interesting. In the Turkish case, there is a pressure and temperature difference (eclogite vs. garnet amphibolite).

If we consider the pressure drop effect, we still need to explain the temperature difference:

*if the 25 kbar of the eclogite is not true and correspond only to 10 kbar (taking the fig. 1 of Yamato and Brun, 2016), the temperature estimate is*

generally solid (~ 500C). Considering the garnet-amphibolite case (~8-10 kbar), things are more difficult. The paper does not provide data below 10 kbar. If we consider that the behavior is purely linear, this implies a pressure drop of 5 kbar. The temperature is ~800C (generally solid). We end up with a DT of 300C, that at a similar pressure will still exists and even increase (if we consider a linear geothermal gradient that is ok at first order). So despite the fact that lithostatic pressure might not be the best practice, the temperature variation is still there

Because our paper specifically addresses lateral temperature differences, we do not include the discussion on the conversion of pressures to depth to our paper, since it doesn't change the point we're making.

Line 27: Please explain in a few words what is supra-subduction, as compared to subduc-tion for the general audience. Supra subduction here is the ophiolite type.

A sentence was added to explain what supra-subduction ophiolite are.

Page 4. Lines 3-7: Please rephrase this sentence. It is too long. The sentence was divided in two parts

Line 10: Nice transition/motivation to the next section. Thanks

Line 15: measurements. *Corrected*

Paragraph 22-33: This paragraph provides some background on previous studies investigating the effect of ge-ometry (obliquity) on subduction dynamics. However, more should be included on stud-ies that look at development of trench curvature (i.e. Schellart et al - convex/concave due to slab width, or sinusoidal when there is both trench advance and retreat), be- cause these studies are more relevant to the present investigation.

A ref and a sentence about the paper by Schellart et al., 2007 is now provided. Thanks for raising that point that we simply have forgotten in the amount of work present in the literature.

Page 5. Line 8: Why no analytical solution in 3D? Perhaps because of its too complex nature (i.e. take into account poloidal and toroidal components and other complex features)? \\

Generally speaking, the Stokes problem has an analytical solution in 2 or 3 D only with very specific boundary conditions (for example the SolCx, SolKz or SolVi benchmarks – passed by our code). The corner flow problem itself has an analytical solution in 2D (Batchelor, 1967, England et al., 2004), but we are not aware of any in 3D. Also, a lot of work was restricted to 2D in previous decades because of computer limi- tations at that time. In principle, 2D is a first order approximation, which yielded some important results, but with some limitations.

The authors could explain why the transi-tion from 2D to 3D studies (i.e. trench obliquity is an inherent 3D feature). The sentence was rephrased accordingly

Line 9: setup. Corrected

Line 23: parameter values . . .Table 1. Added

Line 24: Decoupled energy equation: what about dislocation+diffusion creep, with P,T dependence for mantle viscosity? How would that affect temperature advection on the interface?

The model is isoviscous. Predicting the effect of disl. + diff. creep in such a 3D model might not be trivial, but following the 2D work of van Keken (2002) that would probably increase the temperature at the plate interface.

Line 32 (throughout manuscript): Need to be consistent with units, especially for the time unit (yr): Ma, My, Myr, cm/yr etc. The occurrence of Myr was corrected to My. Velocities are now only expressed as mm/yr and the occurrences of cm/yr were changed. My denotes the age of something without a reference (the age of the plate in the model). Ma relates to an absolute age.

Model Setup: Should indicate before line 30, that that the model setup and boundary conditions are tuned for the Anatolian case study explained in Section 2. *This part was rephrased*

Page 6. Line 7: than.

Corrected

Line 16: use. Done

Line 15-19: Any computational libraries that need to be cited here? *Indeed. The refs were added*

Line 20: How deformed are the Q1Q1 elements to conform with the geometry? Is that affecting the accuracy of the solution? The deformation of the elements is not big, therefore we expect no problems with the accuracy of the solution.

Page 7. Line 9: multiple typos. Corrected

Why use these particular boundary conditions? Is mass con- served? These BC sounds more realistic with respect to previous studies. And of course mass is conserved The inflow/outflow bc are not clearly explained (i.e. flow comes in horizontally from the top right boundary and flows out at the bottom boundary, conserving the mass. . . *This is now better explained in the manuscript*

Line 24: sentence is not clear. *Rephrased*

Line 28, Figure 3: Why does the magnitude of  $\underline{vy}$  max decrease with depth? Is it a consequence of the model setup?  $\backslash \backslash$

Page 8. Line 4 (end): Reference to Figure 4 is incorrect, as that observation was derived from Figure 3. Corrected

Line 6: becomes. *Ok*

Figure 4: insufficiently explained in the text/caption. What are the pink/light blue lines in figure 4a? The figure caption is now properly written

It also seems in Figure 4b, that path 3 is the warmest (compared to 1,2,4) - which corresponds to location theta max? You are right. Going back to the data, the difference is ~0.5C between path 3 and 2 or 4. We think that this difference even if it exist is not really relevant. (There is an extrapolation error introduced when we calculate the Temperature at a given position; i.e. not on the node of the model).

The sentence was modified accordingly: Fig. 4 illustrates that the path in the centre of the model is the coldest and that paths where theta tend to a maximum are the warmest (paths 1 to 4 being within 1 degree at a similar depth; see zoom on Fig. 4

Line 14: What was the subduction rate for the reference model ( $\underline{\sin 20}$  1) compared to these values?

The velocity of this ref. model is 40 mm/yr ( $\underline{cf}$ . fig. 2). This is now stated in the text and was added in the caption of figure 4.

Line 15-16: Why an increase in subduction velocity produces such a polarity change in the temperature variation? This is not explained why that happens/results are not shown. The paragraph is not clear that it refers to the pink/light blue lines in Fig 4. Also, please clarify the differences between the reference model and these additional experiments.

The paragraph was completed together with the caption of figure 4

Line 20: also called inflection points. *This was rephrased*

Page 9.
Line 3: 100 or 110C (as in the Fig5c)?
We wrote "of more than 100C". It is now written of about 110C.

Line 6: centre (purple contours), flow brings colder material from the surface to the centre of the slab. Concave models: I feel their model description is incomplete. The warmest part of the slab is in the center. How warm relative to the edges? What about  $\underline{vy}$  max?

We do not really understand this comment. The concave model is the inverse symmetry of the concave model. Nothing is changing except the shape of the trench. For Model  $-SIN20\setminus_1$ , the warmest part of the model is in the center, with a temperature difference of about 50C at 80 km, as in the reference model (SIN20\\_1). The horizontal velocity shows similar variation than in the ref model (see vy velocity maps on Fig. 5)

Line 16: max theta is in the inflection point. Yes

Line 20: sentence not clear.

The sentence was rephrased: The trench-parallel velocity reach a maximum of 8.0 mm/yr in model ATAN40\\_05, of 1.8 mm/yr in model ATAN05\\_20 and of 3.5 mm/yr in model ATAN10\\_20

Lines 21-22: typos. *corrected*

Line 23: Why is the temperature field asymmetric in Figure  $\underline{6b}$ , slice at 75 km depth, as compared to a,c?

Looking at the data, this comes from the interpolation with the visualization software paraview (the 380C isotherm does not show this asymmetry anymore and the 370 is not present everywhere). The figure was changed with the 380 isotherm that is not confusing.

Line 26, Figure 6: T variation for ATAN0520 (fig 6b) is indicated 75C, not 200C as suggested in the text./ If you mean ATAN1020, the T variation in fig 6c is indicated 110C, even if it looks around 200C. Please revise figure 6 and paragraph. The figure and paragraph were revised

Page 10. Line 9: In which way the results in this paper agree well with previous work? The differences of temperature along strike in other work is now given to emphasis that our calculation agree well with other works.

Line 16: Theta remains constant in the simulations here. Should make it clear and potentially discuss implications for variable Theta (age, velocity, subduction angle)

Theta is not constant here: it varies from 0 to theta max in each simulations

Lines 24-27: The limitations should be extended a bit more. For example, how would power-law T,p dependent rheology of the mantle

expect to influence the T variations? Are the T variations calculated here lower bound estimates?

This is partly discussed: "third, the isoviscous rheology we use is known to underestimates the temperature predicted in the mantle wedge (van Keken et al., 2002)". In their paper, van Keken et al (2002) discussed the effect of isoviscous vs. non-linear (or power law) rheologies. It is difficult to predict if our T variations are lower bound, but considering the increase of velocity variation with non-linear rheologies (Jadamec and Billen 2010), they might.

Line 25: typo. Corrected

Page 11.

Lines 11-12: What are the limitations/improvements of the numerical model that could produce a larger T variation as observed in Anatolia?

Amongst other: our model is ocean/ ocean, the Anatolian case study is continent going under ocean: that would probably change the downgoing plate velocity. There is ophiolite spreading/magmatism in the upper plate of the Anatolian case, so partial melting in response of fluid release could help. The rheologies are probably non linear in the reality.

Figures and Tables.  $\setminus$

Figure 1: Please add some labels/names of the major subduction zones used to illus-trate in the text (i.e. Aleutians, remnants of W-C Turkey). b) The name abbreviations in the figure are not clear. Please either explain in caption or in figure.

The map was completed with names of the subduction zones. We also added the position of the fossil Anatolian subduction zone.

Figure 3: Labels for Fig  $\underline{3g}$ , h, i are missing (as indicated in the text on Page 8, line 2)

The labels g,h,i where part of an early version of the figure. They were removed from the text.

Figure 4: Incomplete figure caption/figure legend. Labels are missing (a,b). What do the paths, vbc, "middle/edge" represent? Some sentences in the caption could help explain the figure better. *Figure caption was completed*

Figure 5: Please indicate simulation labels (i.e. sin20\\_1) in Fig
5a-b. c).
Done\\

The colours for the maximum temperature paths are misleading. If they are all taken at the inflection point (yellow in the sketch), they should all be yellow (like the purple curves). Similarly for the concave model.

Yes, but they are not all taken at the inflection point. (see the  $\underline{3d}$  view with the  $\underline{450C}$  isotherm). The colour of the paths indicates their respective location (as on the inset).

Figure 6:  $\underline{Vy}$  direction arrows as in Figure 5a,b would be useful. Arrows are already present on the Fig. 6 (3 d views). We made them bigger. Below is the marked version of the latex source file of the manuscript. We removed most of the latex commands to make the file more readable. Alexis Plunder and co-authors

**The effect of obliquity on temperature in subduction zones: insights from 3D numerical modeling.**

Alexis Plunder1,a, Cédric Thieulot1 and Douwe J. J. van Hinsbergen1

[1] Department of Earth Sciences, Utrecht University, The Netherlands

[a]{Now at Sorbonne Université, CNRS-INSU, Institut des Sciences de la Terre Paris, ISTEP UMR 7193, F-75005 Paris, France

Running title: Temperature in oblique subduction

abstract

The geotherm in subduction zones is thought to vary as a function of the subduction rate and the age of the subducting lithosphere. Along a single subduction zone the rate of subduction may strongly vary due to changes in the angle between the trench and the plate convergence vector, i.e. the subduction obliquity, due to trench curvature. We currently observe such curvature in e.g. the Marianas, Chile, and the Aleutian trenches. Recently, strong along-strike variations in subduction obliquity were proposed to have cause a major temperature contrast between Cretaceous geological records of Western and Central Turkey. We here test whether first-order temperature variation in subduction zone may be caused by variation of the trench geometry using simple thermo-kinematic finite element 3D numerical models. We prescribe the trench geometry by means of a simple mathematical function and compute the mantle flow in the mantle wedge by solving the equation of mass and momentum conservation. We then solve the energy conservation equation until steady-state is reached. We analyze the results (i) in terms of mantle wedge flow with emphasis on the trench-parallel component, (ii) in terms of temperature along the plate interface by means of maps and depths-temperature path at the interface. In our experiments, the effect of the trench curvature on the geotherm is substantial. A small obliquity yields a small but not negligible trench parallel mantle flow leading to differences of 30 degree C along strike of the model. Advected heat causes such temperature variations (linked to the magnitude of the trench parallel component of velocity). With increasing obliquity, the trench parallel component of the velocity consequently increases and the temperature variation reaches 200\degree C along strike. Finally, we discuss the implication of our simulations for the ubiquitous oblique systems that are observed on Earth and the limitation of our modeling approach. Lateral variations in plate sinking rate associated with curvature will further enhance this temperature contrast. We conclude that the synchronous metamorphic temperature contrast between Central and Western Turkey may well have resulted from reconstructed major variations in subduction obliquity.

**\introduction**

Oceanic subduction and continental collision zones represent approximately 55 000 kilometers of converging plate boundaries on Earth today. They are primarily associated with arc magmatism and seismicity, which in turn are mainly a response to the thermal structure and geotherm of a subduction zone. Numerous studies using 2D high resolution numerical models have addressed the effect of temperature in subduction zones and its link to the coupling of the subduction interface \citep{Wada2009} and related seismicity \citep{Kirby1996,Peacock1999a,Hacker2003a}, as well as the release of fluids \citep{VanKeken2011,Wada2012}, and the associated generation of melt \citep{Gorczyk2007a,Bouilhol2015}. Temperature distributions in subduction zones are thought to vary primarily as a function of the subduction rate and the age of the subducting lithosphere, with lower subduction rates and younger lithosphere tend to increase temperatures at the subduction interface \citep{Kirby1991, Peacock1999a, VanKeken2011}. The geotherm is then mainly controlled by trench perpendicular flow (poloidal) with presumably little variation along-strike. This poloidal flow allows the transport of heat by means of advection (see below).

Real subduction zones, however, tend be curved, i.e. trench strike varies laterally and the angle between the absolute plate motion at the trench and trench strike — the subduction obliquity — thus change along-strike. In fact, some degree of oblique subduction is the rule rather than the exception, both in todays snapshot of plate tectonics (e.g. Fig. \ref{fig\_01} and \cite{Bird2003}) as well as in the geological past \citep{Stampfli2002}, e.g. in the Mediterranean region \citep[e.g.][]{vanhinsbergen2016,Menant2016}, the South-American system \citep{Verard2012,Schepers2017} or the western North American margin \citep{Johnston2001,Liu2008}. \\

Lateral variations in subduction obliquity may conceptually influence the temperature at the subduction interface in two ways. First, oblique subduction adds a component of horizontal relative motion between slab and mantle wedge - toroidal flow - to the poloidal flow in the mantle wedge, which may influence heat advection. Second, higher subduction obliquity leads to a lower net subduction rate. Intuitively, this may suggest that increasing subduction obliquity may be associated with higher temperatures at the subduction interface. Investigating the effect of trench curvature on alongstrike variations of temperature at the plate interface may thus help to explain along-strike variations in e.g. generation of magma, or seismicity along subduction zones, or to help the reconciliation of contrasting metamorphic records with kinematic reconstructions. Few studies were conducted on the effect of obliquity/geometry on the geotherm of subduction zone \citep[e.g.][]{Ji2015} and most studies mainly focused on mantle flow patterns

\citep{Honda2005,Kneller2008,Jadamec2010,Jadamec2012,Bengtson2012,Mor ishige2014,Wada2015}

In this paper, we aim to study the effect of the trench curvature on along-strike temperature distribution changes in subduction zones. In particular, we aimed to test a recent hypothesis that a major, more than 300\degree C along-strike contrast in subduction zone temperature concluded from the geology of Turkey resulted from a corresponding major change in trench strike \citep[see next section;][]{vanhinsbergen2016}.

To this end, a simple 3D thermo-kinematic numerical setup was designed and computed using the finite element code \textsc{elefant} \citep{Thieulot2014,Lavecchia2017}.

Below, we review selected present-day subduction zones and their geometric characteristics as basis for our numerical model setting. Then, we summarize the rationale behind the hypothesis of  $cite{vanhinsbergen2016}$  relating lateral variations in metamorphic grade recognized in the geology of western and central Turkey to oblique subduction. After that, we provide the results from a series of 3D numerical experiments and discuss the limitations of our simple experiments. We evaluate the implications of slab shape or trench geometry on the thermal regime of subduction zone and finally, compare the numerical results with the geological examples of Turkey and the Franciscan complex.

**\section{Oblique subduction: present and past examples}**

Many present-day subduction zones show an along-strike variability in the angle between the absolute motion direction of the downgoing plate and the trench. Fig. \ref{fig\_01} shows that a majority of 100s to >1000 km long subduction zones have concave (e.g. Marianas; Sunda-Burma), or convex (central South America, Northeast Japan) shapes. Some trenches contains as much as 90\degree curvature such that along the same trench, subduction may gradually (Aleutians, Sunda-Burma) or abruptly (southern Marianas, northern Lesser Antilles) change from near-orthogonal subduction to near-transform motion. The subduction rate along such curved subduction zones must change as a function of trench strike. This is best illustrated by the Aleutian trench (Fig.  $ref{fig_01}$ ). In the eastern, NE-SW striking part of the trench, subduction is almost orthogonal, i.e. the plate motion of the downgoing Pacific plate is almost perpendicular to trench strike. In the western, NW-SE striking part of the Aleutian trench, there is almost no subduction and Pacific plate motion is almost parallel to the trench \citep[e.g.][]{ Mccaffrey1992}. This is also reflected in the westward decrease of the length of the Aleutian subducted slab \citep{vandeMeer2017}. Consequently, the subduction rate along the Aleutian trench must gradually decrease from east to west with increasing subduction obliquity. \\

subduction rate is a primary control on the geotherm Τf \citep[e.g.][]{VanKeken2011}, then along-strike variation in
obliquity, should logically lead to along-strike changes in temperature at subduction interfaces. However, determining how strong these lateral variations may be is difficult to estimate from present-day subduction zones due to the lack of proxy to record them. \cite{Plank2009} provided a method to estimate the temperature at the plate interface using melt-inclusions in arc volcanic rocks. Such data suggested along strike variations of temperature exist and can vary through time for example below Central America \citep{Cooper2012}. Better-constrained estimates for the temperature are available for paleo-subduction interfaces through studies of exhumed metamorphosed rocks in subduction-related orogens. These studies demonstrated that the thermal conditions in subduction zones through varied time \citep[e.g.][]{Agard2009,Plunder2015,Angiboust2016}, but also alongstrike. For instance, in the Franciscan complex of California \citep{Wakabayashi2007}, in the Sulawesi m\'elange in SE Asia \citep{Parkinson1996}, in the sub-ophiolitic m\'elanges of Guatemala versus Cuba \citep{Garcia-Casco2007}, (garnet)-amphibolites (high-temperature and mid-pressure condition) are coeval with eclogite or blueschist (low temperature and high-pressure condition) along-strike in the same subduction complex. Taking the pressure (simply assumed to represent depth) of metamorphism into account, these may suggest along-strike temperature differences of ca. 300\degree C. Less dramatic along strike temperature differences at similar depth of (ca. 100\degree C) have also been recorded in Miocene subductionrelated metamorphic rocks of Crete, Greece \citep{Jolivet2010}. \\

An extreme case of along-strike coeval metamorphic temperature variation was reconstructed from the geological record of Turkey. There, two belts of metamorphosed continental rock known as the Tav\c{s}anl{\i } zone and the K\i r\c{s}ehir block experienced coeval metamorphism at strongly contrasting grades during their underthrusting/subduction below oceanic lithosphere that is preserved as ophiolites formed above the nascent subduction zone and are referred as of supra-subduction zone type \citep{
Pearce1984,Dilek1999}. They formed \$\sim\$5-10 Myr before climax metamorphism of the K\i r\c{s}ehir Block and Tav\c{s}anl{\i } zone

Both units were metamorphosed around 80-90 Ma \citep[e.g.][]{Whitney2004,Fornash2016,vanhinsbergen2016,

Pourteau2018} within the same subduction system but under dramatically different metamorphic conditions.

In the Tav\c{s}anl{\i } zone peak metamorphic conditions were estimated to be around 24 \$\pm\$ 2 kbar and 500 \$\pm 50\$\degree C \citep{Okay2002,Plunder2015}, whereas peak metamorphism was estimated around 800 \$\pm\$ 100\degree C and 8 \$\pm\$ 1 kbar in the K\i r\c{s}ehir block \citep{Whitney2004,Lefebvre2015}.

This would suggest that at similar depths, an along-strike temperature variation of more than  $500\$  degree C existed (i.e.  $\$ sim $200\$ degree C for the Tavcsanl{\i } zone at  $\$ sim $25\$  km depths compared to  $800\$ degree C for K\i rcsehir at the same depth).

The paleogeographic transition between the  $\underline{Tav}c{s}anl{i}$  and K\i r\c{s}ehir blocks has been deformed during later continent-continent collision processes, but appears to be abrupt, presently within tens of kilometers (Fig. \ref{fig\_01}b).

Paleogeographic and kinematic reconstructions of Central and Western Anatolia \citep{Lefebvre2013, vanhinsbergen2016,Gurer2016}, suggest that the only major difference between the  $Tav c{s}anl{i}$  and Ki r\c{s}ehir parts of the belt was the angle at which they were buried along the intra-oceanic trench below the oceanic lithosphere now found as ophiolites (Fig. \ref{fig\_01}b). Such reconstruction are constrained based on structural geology and paleomagnetism, and more importantly are independent from interpretations of the causes of the contrast in metamorphism. Subduction of the belt was driven by \$\sim\$NNE-SSW convergence between Africa and Eurasia. The  $\underline{Tav}c{s}anl{i}$  zone was proposed to have been buried by nearorthogonal subduction along an \$\sim\$E-W trending trench segment, whereas the K\i r\c{s}ehir block would have been subducted highly obliquely (Fig. \ref{fig 01}b) along a N-S striking trench segment, which was tentatively proposed to explain the stark metamorphic contrast \citep{vanhinsbergen2016}. Inspired by these geological examples and the hypothesis derived from those, we here aim to perform numerical experiments to test whether, and to what extent, the reconstructed thermal variations may be explained by along-strike variation in subduction geometry.

\section{Model setting}

**\subsection{Background}**

With increasing quality of geophysical measurements and network density, today's tomographic images allow to observe the geometry variation of slab geometry with depths \citep{vandeMeer2017}. Such variations of slab shape are observed below the strait of Gibraltar \citep{Bezada2013}, below Turkey \citep{Biryol2011}, below Japan \citep{Zhao2012,Liu2016}, below the eastern Caribbean plate \citep{VanBenthem2013} and in many other subduction zones, and are summarized in the SLAB1.0 model \citep{Hayes2012}. These complicated pictures of slab geometry allow us to make simple tests to study the possible effects of geometry on the mantle flow and on temperature in subduction zones, and especially at the subduction interface.

Previous 3D thermo-kinematic numerical modeling studies have shown that variation of the geometry of the subduction zone may affect mantle flow patterns and may help to explain seismic anisotropy observed in subduction systems \citep[e.g.][]{Kneller2007}. Numerical models also suggested that the obliquity of subduction zones may have an effect on the temperature at the subduction interface \citep{Bengtson2012,Morishige2014,Ji2015} but did not explore the relationship of such effects with the geological record. These studies have primarily shown that mantle flow may be related to the geometry of the slab edges that lead to the development of toroidal cells \citep[i.e. with trench parallel material transport;][]{Kiraly2017,Schellart2017}. Such trench-parallel, toroidal mantle flow has been proposed as a possible mechanism for differences in volcanic activity along subduction strike \citep{Faccenna2010}.

Some mechanical studies have investigated the effect of trench geometry on the development of topography in the upper plate \citep[e.g.][]{Bonnardot2008}. They also showed that plates bend in relation to the trench shape. \cite{Schellart2007} in their study showed that the shape of a slab is controlled by its original width and evolve in time. Similar studies show that dynamic subduction systems develop 3D geometry with curvature as observed in nature, but in general such models are only mechanical and do not consider temperature \citep{Pusok2015,Kiraly2017,Schellart2017}, or the temperature pattern was not discussed in detail \citep{Jadamec2010,Jadamec2012,Chertova2014, Haynie2017}. Hence, in our study, we aim to test to what extent trench geometry influences the geotherm of a subduction zone.

**\subsection{Numerical rationale and methods}**

The pioneering works of \cite{Batchelor1967} and \cite{McKenzie1969} allowed to investigate the thermal state of subduction zones by providing an analytical solution in 2D, whereby corner flow (i.e. poloidal flow) is dominant. Following these works, many studies were conducted on the behavior of subduction zones using analytical solution \citep{Tovish1978, England2004a} or numerical approximations of corner flow, taking into account stress and temperature dependence of the material in the mantle wedge \citep[e.g.][and references therein]{vankeken2002}. However subduction and particularly the shape of slabs is a 3D problem for which no simple analytical solution exists. To investigate the effect of obliquity on mantle flow and on the temperature at the plate interface, we designed a simple numerical setup using a reference model, and compute deviations from that reference for a set of models in which we vary trench shape. In addition, we briefly test the effect of subduction rate, subduction angle an the downgoing plate age on the thermal state of the plate contact for the reference model. For geological cases where a plate

subducts with an along strike varying obliquity it is then possible to add up the effects of trench geometry and subduction rate on mantle flow and therefore on the temperatures.  $\backslash\backslash$

We used the finite element code \textsc{elefant} \citep{Thieulot2014,Lavecchia2017} to solve the mass, momentum and energy conservation equations in three-dimensions: \\

\begin{eqnarray}
{\bm \nabla}\cdot{\bm v} = 0 \label{eq1} \\
-{\bm \nabla}P + {\bm \nabla} \cdot (2 \mu \dot{\varepsilon}) = 0
\label{eq2}\\
\rho\_0 C\_p \left( \frac{\partial T}{\partial t} + {\bm v}\cdot{\bm
\nabla}T\right) = {\bm \nabla} \cdot (k {\bm \nabla}T) \label{eq3}
\end{eqnarray}

[revised manuscript text omitted]
  $\sinh12\$ ,  $\sinh16\$  and  $\sinh17\$  of the magnitude velocity  $v_{mag} = | bm v |$  at the same location respectively (Fig. \ref{fig 03}d,e,f and Table \ref{table 02}).

In summary, a trench parallel flow develops in the mantle wedge in relation with the shape of the trench. The trench parallel component of the flow cancels at the center of the model and adds up to the along plate interface flow to transport more material at depth.

**\subsubsection{The thermal structure}**

The thermal structure is presented in a 3D view and as map views across the mantle wedge at different depths (Fig.  $ref{fig_03}$ ).

In addition, depth-temperature paths measured along the plate interface are provided but only shown for one half of the model because they are symmetric. As discussed before, the velocity field converges towards the center of the modeling space (Fig.  $ref{fig_03}$ ). Due to advection it appears logical that the 450\degree C isotherm is deflected downward in the center of the modeling space (i.e. where the trench parallel velocity becomes zero; Fig.  $ref{fig_03}c$ ). As a consequence the thermal regime of the subduction zone is different along strike and is cooler in the middle of the domain.

This is also well illustrated with the depth-temperature path along the interface (Fig. \ref{fig\_04}a). Fig. \ref{fig\_04}a shows that the path in the center of the model is the coldest and that paths where theta tends to a maximum are the warmest (paths 1 to 4 being within 1 degree at a similar depth; see zoom on Fig. \ref{fig\_04}). For this reference model the along-strike variation of temperature at the subduction interface is 33\degree C at a depth of 75 km (See inset on Fig. \ref{fig\_04}a). This variation becomes smaller at shallower depth when the subduction interface gets closer to the fixed overriding plate. This region namely the cold nose is even sometimes considered as fixed \citep[e.g.][]{vankeken2002}.

\subsubsection{The effect of slab age, slab velocity and the subduction angle on the thermal structure}

A set of additional experiments with varying slab age, subduction rates and subduction angle was also calculated. We tested the thermal structure at steady state for subducting slab with thermal ages of 50, 75 and 100 My in addition to the reference case where the thermal age is 25 My. The results are shown on Fig. \ref{fig\_04}b. As expected, the thermal regime decreases with the increasing age of the downgoing plate. The temperature spread along strike is not influenced much by the age of the slab and gives temperature difference of **\$\sim\$33\degree** C along strike (Table \**ref**{table\_02}).

With velocities of 20 and 70 mm/yr for the reference concave model (and compared to the 40 mm/yr velocity in the reference experiment) the temperature spread along strike show slight variations (Table  $\ref{table 02}$ ). With 20 mm/yr, the geotherm of the subduction zone gets about 50\degree C warmer (as represented by the red depthtemperature path on Fig.  $\ref{fig_04}$ c with a difference along strike of 34\degree C (Table  $\ref{table 02}$ ). With 70 mm/yr the global geotherm gets 50\degree C colder (blue depth-temperature path on Fig.  $\ref{fig_04}$ c) with a difference along strike of 31\degree C (Table  $\ref{table_02}$ ). The temperature difference along strike is for the three cases largely within the uncertainties of our calculations and we considered that the difference has no proper significance and we consider a difference of ca. 30\degree in each models.

With a subduction angle of 30\degree (\$\alpha\$ on Fig. \ref{fig\_02}), compared to the 45\degree angle in our reference experience, the temperature spread slightly increase along strike and reaches ca. \$\sim\$50\degree C as shown by the depth-temperature curves on fig \ref{fig\_04}c. Again with varying velocities the general thermal regime in decreasing with increasing subduction rate (Fig. \ref{fig\_04}d). All results are summarized in table \ref{table\_02}.

**\subsection{Summary of the other experiments} \subsubsection{Convex and concave models}**

\textbf{Convex models:} The maximum value of \$\theta\$ is measured either at \$\sim42\$ or \$64\$ km corresponding to the so-called inflection point (i.e. where the second derivative of eq. \ref{eq4} equals zero). As in the reference experiment, the mantle wedge flow shows a trench parallel velocity with a increasing value in the region where the obliquity is the highest. The maximum trench parallel velocity is reported in table \ref{table\_03} for all convex experiments. It accounts for up to 49% of the magnitude velocity at 75 kilometer depth in the model with the biggest amplitude (60\$km) and \$\beta = 1\$. When \$\beta = 2\$ and with the maximum amplitude, the trench parallel velocity may even account for 98\% of the total velocity field at a depth of 90 km (Fig. \ref{fig 05}; Table \ref{table\_03}). This trench parallel component of the velocity field is sufficient to allow transportation of heat and creates a symmetric pattern in the temperature field with a colder slab in the middle of the experiment. Our calculations show a difference of temperature of up to 110  $\ensuremath{\mathsf{degree}}$  C for the most extreme configuration tested (i.e. model  $SIN60 \setminus 1$  with  $\theta = 36 \otimes;$  Fig. \ref{fig\_05}a,c; Table \ref{table\_03}).\\

\noindent \textbf{Variation of the curvature:} When varying the wavelength of the curvature of the experiments (e.g. set of experiments SIN40\\_2, SIN60\\_2, or -SIN40\\_2, -SIN60\\_2) the mantle flow pattern and thermal structure also show trench parallel variations. As in the reference model SIN20\\_1, the mantle flow is affected by the shape of the slab and shows some maximum trench velocity perturbation where the obliquity is maximum (e.g. \$\theta = \$17\degree at \$x = 42\$ km and \$v\_y =\$ 12 \% of \$\bm v\$). This leads (1) to a difference in the velocity field, with \$v\_y\$ representing up to 50\% of the total velocity at 75 km depth and (2) to a variation of the plate interface temperature of about \$\sim\$110\degree C (Table \ref{table\_03}). This difference of temperature is observed in a distance of less that a hundred kilometers and is due to the strong trench parallel component of the mantle flow. Interestingly the coldest thermal regime in the convex models does not correspond to the edge of the modeling space where no trench parallel flow is allowed (see boundary condition on Fig. \ref{fig\_02}) but rather to the center. This is due to massive transport of mantle material towards the center of the convex slab. \\

**\subsubsection{S-shaped models}**

The models described hereafter are named ATANA\\_\$\gamma\$ with reference to the parameters of Eq. \ref{eq5}. The maximum obliquity angle (see Eq. \ref{eq5}) is by definition located at the inflection point (i.e. at the center of the model). Its value evolves from 21\degree ~to 38\degree ~in our experiments. As seen in the previous section the shape of the box influences the pattern of the mantle corner flow. For all presented boxes, the mantle flow shows some deviation towards the right as depicted by the white arrows on the 3D view (Fig. \ref{fig 06}: 3D views and \$v y\$ maps) with a maximum

We computed the maximum temperature difference between the center and side of the model to be of about 200\degree C for models ATAN40\\_05 and ATAN10\\_20 (\$\Delta T\_{75 km}\$ equals 200\degree C and 190\degree respectively; Fig. \ref{fig 06}). In model ATAN40\\_05 the shape of the 450\degree C isotherm is relatively smooth whereas in model ATAN10\\_20 it is sharper in direct relation with the shape of the trench. It shows that similar differences of temperature along strike can be obtained with different geometries. The model ATAN05\\_20 has a maximum difference of 75\degree C between the middle and the coldest edge. The step in the shape of the 450\degree C isotherm is minimal. The comparison with model ATAN10\\_20 illustrates that increasing obliquity leads to increasing temperature variations.

**\section{Discussion} \subsection{Implications of obliquity in subduction systems}**

We now evaluate the implications of our results for along-strike temperature variations in subduction zones with obliquity variations that consume a single plate. Our numerical experiments show a straightforward link between mantle wedge flow, the temperature at the plate interface, and the geometry of the subducting slab due to trench shape. This is observed for all type of geometries that we explored (convex, concave, S-shaped). The geometry affects the mantle wedge flow and adds a toroidal flow component to the dominant poloidal flow.

This toroidal flow affects the temperature pattern at the plate interface. The temperature difference may become as much as  $s\sim 200\degree$  C in models with an obliquity of  $s\sim 40\degree$  ~(model ATAN40\ 05 or ATAN10\ 20; Fig. 6).

These results agree well with previous numerical modeling work showing differences of temperature of ca. 100-200\degree C at 90 km depth \citep{Bengtson2012, Morishige2014,Wada2015}, ca. 120-350\degree C depending the depth \citep{Ji2015} or about 50\degree C at the base of the seismogenic zone \citep{ Voshicka2007} Our

at the base of the seismogenic zone \citep{ Yoshioka2007}. Our systematic study of the influence of the shape of the trench on the geotherm shows that a larger amplitude in the model (convex of concave) leads to a larger trench parallel flow and consequently a larger difference in the temperature at the plate interface. The Sshaped model is particularly interesting as it shows that even a small difference in geometry will be expressed as a trench parallel flow and a change of the temperature.

The thermal regime is thought to be controlled by the angle of the subduction and the velocity and age of the downgoing plate, known as the \$\Phi\$ parameter \citep[\$\Phi = AV\_n sin(\delta)\$; with \$A\$ the age of the incoming lithosphere, \$V\_n\$ the normal velocity of the incoming plate and \$\delta\$ the subduction angle;][]{Kirby1991}.

Following our experiments (see Fig. \ref{fig\_04}), \$\Phi\$ remains certainly the first order parameter but we demonstrate that the trench parallel mantle flow influences on the temperature at the plate interface and may thus explain along-strike temperature differences in subduction zones. A 2D approach remains viable in systems with small obliquity, as stated in \cite{Bengtson2012}, but important variations of geometry should be considered in further studies to reliably represent subduction zone dynamics both for present-day and past systems.

**\subsection{Limitations}**

The experiments with an amplitude variation of 20, 40 or 60 km over 250 km display a substantial effect on the mantle wedge flow and temperature pattern at the plate interface. Such geometrical variations are observed on Earth (e.g. in the Andean subduction around the border between Chile and Peru, or below Japan (Fig. \ref{fig\_01})). However, our modelling approach is not without limitations: first, our model setting is a highly simplified geometry of a subduction zone; second, in our kinematic approach the deformation and thus shear heating (especially at the interface) is not taken in account; third, the isoviscous rheology we use is known to underestimates the temperature predicted in the mantle wedge \citep{vankeken2002}. The hypothesis of isoviscosity also reduces the magnitude of the calculated trench parallel flow of at least one order of magnitude compared to a non-linear rheology for the mantle \citep{ Kneller2008, Jadamec2012}. This has naturally an effect on the temperature calculated in our models and allows us only to give only lower bounds estimates for the temperature variation at the plate interface.

In addition no feedback mechanisms induced by effects of temperature change along the plate interface, such as water transport or melting processes that may influence the mechanical behavior, were taken in account. We made these simplifications because it allows for a dramatically lower computation time and was useful to evaluate the qualitative effect of the geometry on the thermal regime.

We primarily aimed to test whether the major contemporaneous alongstrike changes in temperature in geological records of subduction zones may be to first order explained by subduction obliquity changes, and our results suggest that they may indeed.

Our study may thus form the basis for more detailed studies on the effect of obliquity on e.g. dehydration reaction and seismicity in subduction zones as function of obliquity, taking the effects of above limitations into account.

**\subsection{Comparison with the geological record}**

We now compare our models to the geologically constrained temperature variations in paleo-subduction zones. Our study was largely motivated by the geological record from Western and Central Turkey, but as mentioned, similar along-strike temperature variations have been recovered from other geological settings such as the Fanciscan complex \citep[see review by][]{Wakabayashi2015,Wakabayashi2017}, the Sulawesi m\'elange \citep{Parkinson1996}, or the peri-Caribbean m\'elanges \citep{Garcia-Casco2007}. 
[revised manuscript text omitted]

\begin{figure\*}[h] \vspace\*{2mm} \begin{center} \includegraphics[width=1\textwidth]{figs/fig\_01.png} \end{center} \caption{(a) Plate motion at trenches from the NNR-MORVEL model \citep{Argus2011}. Baselayer obtain with GeoMapApp (http://www.geomapapp.org) with topography and bathymetry from \cite{Ryan2009}. Abbreviations as follow: Am., Andaman: As., Alaska; Al., Aleutians; At., Antilles; Ca., Central America; C., Chile; Co. Colombia; Cs., Cascadia; Ge., Greece; H., Hikurangi; In., Izu-Bonin; Nh., Japan; K., Kamchatka; Kr., Kurile; Mn., Marianas; Mo., Mexico; Nb., New Britain; Pa., Palau; Pr., Peru; P., Philippine; S. Sumatra; <mark>Sc. Scotia; Sn., Sunda; Ta., Tonga .</mark> (b) Possible palaeogeographic configuration at ca. 90 Ma for Central Turkey based on the reconstruction of \cite{vanhinsbergen2016}.
Abbreviation correspond to the following units: Af.Ör., Afyon-Ören zone; I.T.B. Inner Tauride Basin; Kır., Kırşehir block; Tav., Tavşanlı zone. **\$Z\_K\$** and **\$Z\_T\$** refer to the maximum burial of the Kirsehit and Tavşanlı units as discussed in text.} \label{fig\_01} \end{figure\*}

\begin{figure\*}[h] \vspace\*{2mm} \begin{center} \includegraphics[width=1\textwidth]{figs/fig\_02.png} \end{center} \caption{Setting of the computed models with the kinematic boundary condition. Initial thermal state is computed following the half-space cooling model following the formulation of \cite{turcotte-schubert} with a 25 My old oceanic lithosphere for the downgoing slab and a 5 My old lithosphere for the upper plate. The physical dimensions are identical for each model and are specified on the convex setting. The number of elements is  $65 \times 10^{10}$  B5  $\pm 10^{10}$  B s  $35^{10}$  The x,y and \$z\$ direction respectively leading to a physical resolution of \$2.3 \times 3 \times 2.30\$ km for each models. } \label{fig\_02} **\end**{figure\*} \begin{figure\*}[h] \vspace\*{2mm} \begin{center} \includegraphics[height=0.88\textheight]{figs/fig 03.png} \end{center}  $\operatorname{Caption}\{\operatorname{Results} \text{ of the model } \underline{\operatorname{SIN20}}\_1.$  (a) Top view with streamlines showing the trench parallel mantle flow and sections  $\bm v$  and  $v_y$  at  $y=64 \ km$ ; (b) rear view of the domain with emphasis on the trench parallel flow represented as stramlines; (c) Temperature pattern in the model and deflection of the 450\degree C isotherm; (d)  $\boldsymbol{v}$  and  $\boldsymbol{T}$  at 60 km depth; (e and f) same as (d) at 75 and 90 km depth. } \label{fig 03} \end{figure\*}

```
\begin{figure}[t]
\vspace*{2mm}
\begin{center}
\includegraphics[width=0.8\textwidth]{figs/fig 04.png}
\end{center}
\caption{(a) Depth--Temperature path retrieved at the plate interface
of the reference model (SIN20\_1) with a downgoing plate velocity of
40 mm/yr. The relative position of the depth-temperature path is
given (middle and edge). The zoom allows to better evaluating the
relative position of the depth-temperature paths and the differences
of thermal regime along-strike. The inset sketch gives the position
of the sampled point along the slab with respect to depth (the color
shading depict the deepening); (b) Variation of thermal regime with
respect to slab age. The blue, green, yellow and orange curves give
the temperature range in the model (ca. 30\ degree C); (c) investigation of different subducting rates for the reference experiment; blue is 70 mm/yr and red is 20 mm/yr; (d) variation of temperature for a subduction angle of 30\
different subduction rates.}
\label{fig_04}
\end{figure}
\begin{figure*}
\vspace*{2mm}
\begin{center}
\includegraphics[height=0.88\textheight]{figs/fig_05.png}
\end{center}
\caption{(a) Mantle flow, shape of the 450\degree C isotherm and
depth temperature path for the convex models (SIN20\_1, SIN40\_1 and
SIN60\_1); (b) Mantle flow, shape of the 450\degree C isotherm and
depth temperature path for the concave models (SIN20\1, SIN40\1 and
SIN60\ 1). Velocity map of the $y$ component are reported at 60, 75
and 90 km depths for the case -sSIN20\_1. (c) summary of the depth-
```

temperature path calculated for the convex and concave models, showing a maximum temperature variation of ca. 110\degree C with the most oblique models. Details of the \$\Delta T\_{75km}\$ are given in Table \ref{table\_02}.}

```
\label{fig_05}
\end{figure*}
```

\begin{figure}
\vspace\*{2mm}
\begin{center}
\includegraphics[height=0.88\textheight]{figs/fig\_06.png}
\end{center}
\caption{
From to to bottom: 3D view of model ATANA\\_\$\gamma\gamma\$ showing the
mantle flow streamlines and the contour of the 450\degree C isotherm
and its more or less important deflection at the center of the
modeling space. The white arrows emphasize the direction of the
mantle flow. They are dimensionless. Map of temperature at 75 km with
isotherms in white. Map of trench parallel velocity at 75 km depth.
Depth--temperature path along the plate interface showing a \$\Delta

```
T$ of 200 \degree C, 75 \degree C or 190 \degree C depending on the
model.
(a) Model ATAN40 \ 05;
(b) Model ATAN05\ 20;
(c) Model ATAN10\_20
\label{fig 06}
\end{figure}
\pagebreak
%%%%%% table 1
8t.
\begin{table}
\caption{Physical parameter used in the numerical model}
\label{table_01}
\vskip4mm
\centering
\begin{tabular}{llrl}
\tophline
Symbol & Name & Value & Units \\
\middlehline

      $C_p$
      & specific heat & $1250$
      & $J.kg^{-1}.K^{-1}$\\

      $\rho$
      & volumetric mass density & $3300$
      & $kg.m^{-3}$\\

$\mu$ & effective viscosity & $10^{22}$ & $Pa.s$\\
$k$
          & thermal conductivity & $2.5 $ & $W.m^{-1}.K^{-1}$
                                                                         \ \
\bottomhline
\end{tabular}
\end{table}
%%%%%% table 2
8t.
\begin{table}
\caption{Variation of temperature along strike for the reference
model (SIN20 \setminus 1) and those derived from it .}
\label{table_02}
\vskip4mm
\centering
\begin{tabular}{lcrrr}
\tophline
Model Name
                       $v_{bc}$ &
                                             $∖alpha$
              &
                                                               &
                       & $\Delta T_{75 km}$ \\
       Slab age
Model Name &
                        ($mm/yr$)
                                      8
                                                               8
       (My)
               &
                        \degree C\\
\tophline
SIN20 \ 1
                       40
                                       45
                                                       25
               &
                               &
                                               &
                                                               &
       33\\
\underline{SIN40} \ 1
                       40
                               &
                                       45
                                               &
                                                       25
                                                               &
               &
       80\\
SIN60\ 1
                       40
                               &
                                       45
                                               &
                                                       25
                                                               &
               &
       110\\
\middlehline
                      40
                                       45
                                                       25
SIN20\ 2
                              &
                                              &
               &
                                                               8
       40\\
```

| SIN40\_2       | &    | 40 | & | 45 | & | 25  | & |
|----------------|------|----|---|----|---|-----|---|
| 91\\           |      |    |   |    |   |     |   |
| SIN60\_2       | &    | 40 | & | 45 | & | 25  | & |
| 143\\          |      |    |   |    |   |     |   |
| \middlehline   |      |    |   |    |   |     |   |
| $30dSIN20 \ 1$ | &    | 40 | & | 30 | & | 25  | & |
| 50\\           |      |    |   |    |   |     |   |
| $30dSIN40 \ 1$ | &    | 40 | & | 30 | & | 25  | & |
| 100\\          |      |    |   |    |   |     |   |
| $30dSIN60 \ 1$ | &    | 40 | & | 30 | & | 25  | & |
| 160\\          |      |    |   |    |   |     |   |
| \middlehline   |      |    |   |    |   |     |   |
| 18mm30dSIN20   | _1 & | 18 | & | 30 | & | 25  | & |
| 50\\           |      |    |   |    |   |     |   |
| 44mm30dSIN20   | _1 & | 44 | & | 30 | & | 25  | & |
| 50\\           |      |    |   |    |   |     |   |
| 56mm30dSIN20   | _1 & | 56 | & | 30 | & | 25  | & |
| 50\\           |      |    |   |    |   |     |   |
| \middlehline   |      |    |   |    |   |     |   |
| 50SIN20\_1     | &    | 40 | & | 45 | & | 50  | & |
| 33//           |      |    |   |    |   |     |   |
| 75SIN20\_1     | &    | 40 | & | 45 | & | 75  | & |
| 33//           |      |    |   |    |   |     |   |
| 100SIN20\_1    | &    | 40 | & | 45 | & | 100 | & |
| 33//           |      |    |   |    |   |     |   |
| \middlehline   |      |    |   |    |   |     |   |
| 70mmSIN20\_1   | &    | 70 | & | 45 | & | 25  | & |
| 31\\           |      |    |   |    |   |     |   |
| 20mmSIN20\_1   | &    | 20 | & | 45 | & | 25  | & |
| 34\\           |      |    |   |    |   |     |   |
| \bottomhline   |      |    |   |    |   |     |   |

**\end**{tabular}

**\end**{table}

%%%%%% table 3

**۶t \begin{table} $caption{Variation of <math>v_y$ in the mantle at different depth compared with the magnitude velocity at the same position. The variation of temperature at 75 km depth is also give to complete table \ref{table\_02}} \label{table 03} \vskip4mm \centering \begin{tabular}{lrr|crr|r} \tophline Model & \$\Delta T {75 km}\$ & depth & \$v y \$ 8 $\setminus \setminus$ & \$ v\$ & & \degree C & \$km\$ & & name δ \$mm/yr\$ & \$v\_y / v\$ \\ & & 90 & 0.944 & 4.54 & 17\% \\ \middlehline SIN20\\_2 & & & & & 60 & & 1.51 & & 15.0 & & 10\% \\ \$\theta\_{max}\$ & 17\degree & 40 & 75 & 1.28 & & 10.5 & 12\% \\ \$y\_{\theta\_{max}}\$ & \$\pm42\$ & & 90 & 0.976 & 4.13 & 24\% \\ \middlehline**

 $SIN40 \ 1$
 SIN40\\_1
 &
 &
 &
 60
 &
 3.08
 &
 10.4
 &
 23\%
 \\

 \$\theta\_{max}\$
 &
 26\degree
 &
 80
 &
 75
 &
 2.54
 &
 8.30
 &
 30\%
 \\
 & & 60 & 3.08 & 10.4 & 23\% \\ \$y\_{\theta\_{max}}\$ & \$\pm64\$ & & 90 & 1.88 & 4.73 & 39\% \\ \middlehline & & 60 & 3.26 & 15.9 & 21\% \\  $SIN40 \ge 2$ 8 \$\theta\_{max}\$ & 32\degree & 91 & 75 & 2.76 & 11.0 & 25\% \\ \$y\_{\theta\_{max}}\$ & \$\pm42\$ & & 90 & 2.14 & 3.79 & 56\% \\ \middlehline  $SIN60 \setminus 1$ & & 60 & 4.68 & 12.3 & 38\% & \$\theta {max}\$ & 36\degree & 110 & 75 & 3.81 & 7.76 & 49\% \\ \$y\_{\theta\_{max}}\$ & \$\pm64\$ & & 90 & 2.84 & 4.78 & 59\% \\ \middlehline SIN60\ 2 & & 60 & 5.37 & 16.2 & 33\% \\ & \$\theta {max}\$ & 43\degree & 143& 75 & 4.58 & 10.9 & 42\% \\ \$y\_{\theta\_{max}}\$ & \$\pm42\$ & & 90 & 3.59 & 3.67 & 98\% \\ \middlehline  $ATAN40 \ge 05$ & & 60 & 8.65 & 10.3 & 84\% \\ & \$\theta\_{max}\$ & 38\degree & 200 & 75 & 8.04 & 9.68 & 83\% \\ \$y\_{\theta\_{max}}\$ & \$128\$ & & & 0 & 6.54 & 6.87 & 95\% \\ \middlehline & & & 60 & 1.88 & 13.0 & 14\% \\ ATAN05\\_20

 ATANOS
 20
 a
 a
 a
 a
 a
 a
 b
 a
 1.00
 a
 10.0
 a
 11.00
 \$y\_{\theta\_{max}}\$ & \$128\$ & & 90 & 1.49 & 4.01 & 37\% \\ \middlehline

 ATAN10\\_20
 &
 &
 &
 60 & 3.74
 &
 13.8 & 27\%
 \\

 \$\theta\_{max}\$
 &
 27\degree
 &
 190 & 75 & 3.50
 &
 10.0 & 35\%
 \\

 \$y\_{\theta\_{max}}\$
 &
 \$128\$
 &
 90 & 2.98
 3.79 & 79\%
 \\

\bottomhline
\end{tabular}
\end{table}

\addtocounter{figure}{-1}\renewcommand{\thefigure}{\arabic{figure}a}

**\end**{document}